# COMPETING LARGE LANGUAGE MODELS IN MULTI-AGENT GAMING ENVIRONMENTS

**Jen-tse Huang**[1,2]    **Eric John Li**[1]    **Man Ho Lam**[1]    **Tian Liang**[5,2]    **Wenxuan Wang**[3,2*]
**Youliang Yuan**[4,2]    **Wenxiang Jiao**[2*]    **Xing Wang**[2]    **Zhaopeng Tu**[2]    **Michael R. Lyu**[1]
[1]The Chinese University of Hong Kong    [2]Tencent AI Lab    [3]Renmin University of China
[4]The Chinese University of Hong Kong, Shenzhen    [5]Tsinghua University

## ABSTRACT

Decision-making is a complex process requiring diverse abilities, making it an excellent framework for evaluating Large Language Models (LLMs). Researchers have examined LLMs' decision-making through the lens of *Game Theory*. However, existing evaluation mainly focus on two-player scenarios where an LLM competes against another. Additionally, previous benchmarks suffer from test set leakage due to their static design. We introduce GAMA($\gamma$)-Bench, a new framework for evaluating LLMs' Gaming Ability in Multi-Agent environments. It includes eight classical game theory scenarios and a dynamic scoring scheme specially designed to quantitatively assess LLMs' performance. $\gamma$-Bench allows flexible game settings and adapts the scoring system to different game parameters, enabling comprehensive evaluation of robustness, generalizability, and strategies for improvement. Our results indicate that GPT-3.5 demonstrates strong robustness but limited generalizability, which can be enhanced using methods like Chain-of-Thought. We also evaluate 13 LLMs from 6 model families, including GPT-3.5, GPT-4, Gemini, LLaMA-3.1, Mixtral, and Qwen-2. Gemini-1.5-Pro outperforms others, scoring of 69.8 out of 100, followed by LLaMA-3.1-70B (65.9) and Mixtral-8x22B (62.4). Our code and experimental results are publicly available at `https://github.com/CUHK-ARISE/GAMABench`.

## 1 INTRODUCTION

We have recently witnessed the advancements in artificial intelligence made by Large Language Models (LLMs), which have marked a significant breakthrough in the field. ChatGPT[1], a leading LLM, has demonstrated its proficiency in a variety of natural language processing tasks, including machine translation (Jiao et al., 2023), sentence revision (Wu et al., 2023), information retrieval (Zhu et al., 2023), and program repair (Surameery & Shakor, 2023). Beyond the academic sphere, LLMs have entered diverse aspects of our everyday life, such as education (Baidoo-Anu & Ansah, 2023), legal service (Guha et al., 2023), product design (Lanzi & Loiacono, 2023), and healthcare (Johnson et al., 2023). Given their extensive capabilities, evaluating LLMs demands more than simple, isolated tasks. A comprehensive and multifaceted approach is highly in demand to assess the efficacy of these advanced models.

With the broad knowledge encoded in LLMs, their intelligence (Liang et al., 2024), and capabilities in general-purpose task solving (Qin et al., 2023), a question emerges: *Can LLMs assist in everyday decision-making?* Many real-world decision-making scenarios can be modeled using *Game Theory* (Koller & Pfeffer, 1997). Furthermore, individuals' ability to achieve Nash equilibrium (Nash, 1950) reflects their capacity in decision-making (Risse, 2000). Therefore, many studies have drawn on the principles of game theory (Duan et al., 2024; Xie et al., 2024; Xu et al., 2024a), which has several advantages: (1) Scope: Game theory allows for the abstraction of diverse real-life scenarios into simple mathematical models, facilitating a broad range of evaluations. (2) Quantifiability: By examining the Nash equilibrium within these models, we gain a measurable metric for comparing

---

*Wenxiang Jiao and Wenxuan Wang are the corresponding authors.
[1]`https://chat.openai.com/`

Table 1: Performance (scores) of different LLMs on $\gamma$-Bench.

| $\gamma$-Bench Leaderboard | GPT-3.5 | | | GPT-4 | | Gemini-Pro | |
| --- | --- | --- | --- | --- | --- | --- | --- |
| | 0613 | 1106 | 0125 | t-0125 | o-0806 | 1.0 | 1.5 |
| Guess 2/3 of the Average | $41.4_{\pm0.5}$ | $68.5_{\pm0.5}$ | $63.4_{\pm3.4}$ | $91.6_{\pm0.6}$ | $94.3_{\pm0.6}$ | $77.3_{\pm6.2}$ | $95.4_{\pm0.5}$ |
| El Farol Bar | $74.8_{\pm4.5}$ | $64.3_{\pm3.1}$ | $68.7_{\pm2.7}$ | $23.0_{\pm8.0}$ | $70.0_{\pm22.1}$ | $33.5_{\pm10.3}$ | $37.2_{\pm4.2}$ |
| Divide the Dollar | $42.4_{\pm7.7}$ | $70.3_{\pm3.3}$ | $68.6_{\pm2.4}$ | $98.1_{\pm1.9}$ | $95.2_{\pm0.7}$ | $77.6_{\pm3.6}$ | $93.8_{\pm0.3}$ |
| Public Goods Game | $17.7_{\pm1.7}$ | $43.5_{\pm12.6}$ | $38.9_{\pm8.1}$ | $89.2_{\pm1.8}$ | $90.9_{\pm3.0}$ | $68.5_{\pm7.6}$ | $100.0_{\pm0.0}$ |
| Diner's Dilemma | $67.0_{\pm4.9}$ | $1.4_{\pm1.3}$ | $2.8_{\pm2.8}$ | $0.9_{\pm0.7}$ | $10.7_{\pm8.3}$ | $3.1_{\pm1.5}$ | $35.9_{\pm5.3}$ |
| Sealed-Bid Auction | $10.3_{\pm0.2}$ | $7.6_{\pm1.8}$ | $13.0_{\pm1.5}$ | $24.2_{\pm1.1}$ | $20.8_{\pm3.2}$ | $31.6_{\pm12.2}$ | $26.9_{\pm9.4}$ |
| Battle Royale | $19.5_{\pm7.7}$ | $35.7_{\pm6.8}$ | $28.6_{\pm11.0}$ | $86.8_{\pm9.7}$ | $67.3_{\pm14.8}$ | $16.5_{\pm6.9}$ | $81.3_{\pm7.7}$ |
| Pirate Game | $68.4_{\pm19.9}$ | $69.5_{\pm14.6}$ | $71.6_{\pm7.7}$ | $85.4_{\pm8.7}$ | $84.4_{\pm6.7}$ | $57.4_{\pm14.3}$ | $87.9_{\pm5.6}$ |
| **Overall** | $42.7_{\pm2.0}$ | $45.1_{\pm1.6}$ | $44.4_{\pm2.1}$ | $62.4_{\pm2.7}$ | $66.7_{\pm4.7}$ | $45.7_{\pm3.4}$ | $69.8_{\pm1.6}$ |

(a) Closed-source LLMs: Gemini-1.5-Pro outperforms. For GPT-4: t denotes Turbo and o denotes Omni.

| $\gamma$-Bench Leaderboard | LLaMA-3.1 | | | Mixtral | | Qwen-2 |
| --- | --- | --- | --- | --- | --- | --- |
| | 8B | 70B | 405B | 8x7B | 8x22B | 72B |
| Guess 2/3 of the Average | $85.5_{\pm3.0}$ | $84.0_{\pm1.7}$ | $94.3_{\pm0.6}$ | $91.8_{\pm0.4}$ | $83.6_{\pm4.6}$ | $93.2_{\pm1.3}$ |
| El Farol Bar | $75.7_{\pm2.2}$ | $59.7_{\pm3.5}$ | $20.5_{\pm24.2}$ | $66.8_{\pm5.8}$ | $39.3_{\pm12.2}$ | $17.0_{\pm25.5}$ |
| Divide the Dollar | $56.4_{\pm8.4}$ | $87.0_{\pm4.1}$ | $94.9_{\pm1.0}$ | $1.2_{\pm2.8}$ | $79.0_{\pm9.6}$ | $91.9_{\pm2.4}$ |
| Public Goods Game | $19.6_{\pm1.0}$ | $90.6_{\pm3.6}$ | $97.0_{\pm0.8}$ | $27.6_{\pm11.7}$ | $83.7_{\pm3.5}$ | $81.3_{\pm5.9}$ |
| Diner's Dilemma | $59.3_{\pm2.4}$ | $48.1_{\pm5.7}$ | $14.4_{\pm4.5}$ | $76.4_{\pm7.1}$ | $79.9_{\pm5.8}$ | $0.0_{\pm0.0}$ |
| Sealed-Bid Auction | $37.1_{\pm3.1}$ | $15.7_{\pm2.7}$ | $14.7_{\pm3.2}$ | $3.1_{\pm1.6}$ | $13.2_{\pm3.7}$ | $2.5_{\pm0.7}$ |
| Battle Royale | $35.9_{\pm12.1}$ | $77.7_{\pm26.0}$ | $92.7_{\pm10.1}$ | $12.6_{\pm9.4}$ | $36.0_{\pm21.0}$ | $81.7_{\pm9.6}$ |
| Pirate Game | $78.3_{\pm10.0}$ | $64.0_{\pm15.5}$ | $65.6_{\pm22.3}$ | $67.3_{\pm7.6}$ | $84.3_{\pm8.8}$ | $86.1_{\pm6.4}$ |
| **Overall** | $56.0_{\pm3.1}$ | $65.9_{\pm3.3}$ | $61.8_{\pm4.7}$ | $43.4_{\pm2.2}$ | $62.4_{\pm2.2}$ | $56.7_{\pm3.4}$ |

(b) Open-source LLMs: LLaMA-3.1-70B outperforms.

LLMs' decision-making performance. (3) Variability: The adjustable parameters of these models enable the creation of variant scenarios, enhancing the diversity and robustness of our assessments. However, existing research is often limited to two-player or two-action settings, such as the classical Prisoner's Dilemma and Ultimatum Game (Guo, 2023; Phelps & Russell, 2023; Akata et al., 2023; Aher et al., 2023; Brookins & DeBacker, 2024). Moreover, prior work relies on fixed, classical game settings, increasing the likelihood that LLMs have encountered these scenarios during training, facing the risk of test set leakage. In this paper, we assess LLMs in more complex scenarios involving multiple players, actions, and rounds, across classical game theory scenarios with dynamically adjustable game parameters.

We include eights games and divide them into three categories based on their characteristics. The first category in our framework evaluates LLMs' ability to make optimal decisions by understanding game rules and recognizing patterns in other players' behavior. A distinctive characteristic of these games is that individual players cannot achieve higher gains without cooperation, provided that other participants cooperate. Essentially, these games' Nash equilibrium aligns with maximizing overall social welfare. We name such games as **I. Cooperative Games**, including (1) *Guess 2/3 of the Average*, (2) *El Farol Bar*, and (3) *Divide the Dollar*. The second category assesses the propensity of LLMs to prioritize self-interest, potentially betraying others for greater gains. In contrast to the first category, games in this category incentivize higher rewards for participants who betray their cooperative counterparts. Typically, the Nash equilibrium in these games leads to reduced social welfare. This category is termed **II. Betraying Games**, including (4) *Public Goods Game*, (5) *Diner's Dilemma*, (6) *Sealed-Bid Auction*. Last but not least, we focus specifically on two games characterized by sequential decision-making processes, distinguishing them from the previous six games based on simultaneous decision-making. **III. Sequential Games** are the (7) *Battle Royale* and (8) *Pirate Game*.

Decision-making is a complex task requiring various abilities. Several common ones are evaluated across all games: (1) Perception: the ability to understand situations, environments, and rules, and extends to long-text understanding for LLMs. (2) Arithmetic Reasoning: the ability to quantify real-world options and perform calculations. (3) ToM Reasoning: the Theory of Mind (Kosinski, 2024; Bubeck et al., 2023; Huang et al., 2024a) refers to the ability to infer others' intentions and

beliefs. (4) Strategic Reasoning: the ability to integrate all available information to arrive at the best decision. Certain games involve specialized abilities, such as K-level reasoning in the "Guess 2/3 of the Average" game and mixed strategy adoption in the "El Farol Bar" game.

In this paper, we instruct ten agents, based on the GPT-3.5 (0125) model, to engage in the eight games, followed by an analysis of the results obtained. Subsequently, we assess the model's robustness against multiple runs, temperature parameter alterations, and prompt template variations. Further exploration is conducted to ascertain if instructional prompts, such as Chain-of-Thought (CoT) (Kojima et al., 2022), enhance the model's decision-making capabilities. Additionally, the model's capacity to generalize across diverse game settings is examined. Finally, we evaluate the performance of **thirteen** LLMs, including GPT-3.5-Turbo (0613, 1106, 0125) (OpenAI, 2022), GPT-4 (Turbo-0125, 4o-0806) (OpenAI, 2023), Gemini-1.0-Pro (Pichai & Hassabis, 2023), Gemini-1.5-Pro (Pichai & Hassabis, 2024), LLaMA-3.1 (8B, 70B, 405B) (Dubey et al., 2024), Mixtral (8x7B, 8x22B) (Jiang et al., 2024), and Qwen-2-72B (Yang et al., 2024). We compare the performance of different LLMs by creating multiple agents from the same model to participate in the games, then calculate the average performance of these agents. Our contributions include:

- We provide a comprehensive review and comparison of existing literature on evaluating LLMs using game theory scenarios, as summarized in Table 3. The review includes key aspects such as models, games, temperature settings, and other game parameters, highlighting our emphasis on the multi-player setting and the generalizability of LLMs.

- Starting from the multi-player setting, we collect eight classical game theory scenarios to measure LLMs' $\underline{G}$aming $\underline{A}$bility in $\underline{M}$ulti-$\underline{A}$gent environments, and implement our framework, GAMA($\gamma$)-Bench. It enables dynamic game scene generation with diverse profiles, offering unlimited scenarios to assess LLM generalizability while minimizing test set leakage risk.

- We apply $\gamma$-Bench to thirteen LLMs to provide an in-depth analysis of their performance in multi-agent gaming scenarios, indicating their potential as assistants in decision-making process.

## 2 INTRODUCTION TO GAMES

We collect eight games well studied in Game Theory and propose $\gamma$-Bench, a framework with multi-player, multi-round, and multi-action settings. Notably, $\gamma$-Bench allows the simultaneous participation of both LLMs and humans, enabling us to evaluate LLMs' performance when playing against humans or fixed strategies. This section details each game with their classical settings (parameters).

### 2.1 COOPERATIVE GAMES

**(1) Guess 2/3 of the Average** Initially introduced by Ledoux (1981), the game involves players independently selecting an integer between 0 and 100 (inclusive). The winner is the player(s) choosing the number closest to two-thirds of the group's average. A typical initial strategy might lead players to assume an average of 50, suggesting a winning number around $50 \times \frac{2}{3} \approx 33$. However, if all participants adopt this reasoning, the average shifts to 33, thereby altering the winning number to approximately 22. The game has a Pure Strategy Nash Equilibrium (PSNE) where all players selecting zero results in a collective win.

**(2) El Farol Bar** Proposed by Arthur (1994) and Huberman (1988), this game requires players to decide to either visit a bar for entertainment or stay home without communication. The bar, however, has a limited capacity and can only accommodate part of the population. In a classical scenario, the bar becomes overcrowded and less enjoyable if more than 60% of the population decides to go there. Conversely, if 60% or fewer people are present, the experience is more enjoyable than staying home. Imagine that if everyone adopts the same pure strategy, *i.e.*, either everyone going to the bar or everyone staying home, then the social welfare is not maximized. Notably, the game lacks a PSNE but presents an Mixed Strategy Nash Equilibrium (MSNE), where the optimal strategy involves going to the bar with a 60% probability and staying home with a 40% probability.

**(3) Divide the Dollar** Firstly mentioned in Shapley & Shubik (1969), the game involves two players independently bidding up to 100 cents for a dollar. Ashlock & Greenwood (2016) further generalized the game into a multi-player setting. If the sum of bids is at most one dollar, each player

is awarded their respective bid; if the total exceeds a dollar, no player receives anything. The NE of this game occurs when each player bids exactly $\frac{100}{N}$ cents.

## 2.2 BETRAYING GAMES

**(4) Public Goods Game**   Studied since the early 1950s (Samuelson, 1954), the game requires $N$ players to secretly decide how many of their private tokens to contribute to a public pot. The tokens in the pot are then multiplied by a factor $R$ ($1 < R < N$), and the resulting "public good" is evenly distributed among all players. Players retain any tokens they do not contribute. A simple calculation reveals that for each token a player contributes, their net gain is $\frac{R}{N} - 1$, which is less than zero. This suggests that the rational strategy for each player is to contribute no tokens, which reaches an NE of this game. The game serves as a tool to investigate tendencies towards selfish behavior and free-riding among participants.

**(5) Diner's Dilemma**   This game is the multi-player variant of the *Prisoner's Dilemma* (Glance & Huberman, 1994). The game involves $N$ players dining together, with their decision to split all the costs. Each player needs to independently choose whether to order the expensive or the cheap dish, priced at $x$ and $y$ ($x > y$), respectively. The expensive offers $a$ utility per individual, surpassing the $b$ utility of another choice ($a > b$). The game satisfies two assumptions: (1) $a - x < b - y$: Although the expensive dish provides a greater utility, the benefit does not justify its higher cost, leading to a preference for the cheap one when dining alone. (2) $a - \frac{x}{N} > b - \frac{y}{N}$: Individuals are inclined to choose the expensive dish when the cost is shared among all diners. The assumptions lead to an NE where all players opt for the more expensive meal. However, this PSNE results in a lower total social welfare of $N(a - x)$ compared to $N(b - y)$, which is the utility if all choose the cheap one. This game evaluates the long-term perspective and the capacity to establish sustained cooperation.

**(6) Sealed-Bid Auction**   The *Sealed-Bid Auction* (SBA) involves players submitting their bids confidentially and simultaneously, different from the auctions where bids are made openly in a sequential manner. We consider two variants of SBA: the *First-Price Sealed-Bid Auction* (FPSBA) and the *Second-Price Sealed-Bid Auction* (SPSBA). In FPSBA, also known as the *Blind Auction*, if all players bid their true valuation $v_i$ of the item, the winner achieves a net gain of $b_i - v_i = 0$ while others also gain nothing (McAfee & McMillan, 1987). Moreover, the highest bidder will discover that to win the auction, it is sufficient to bid marginally above the second-highest bid. Driven by these two factors, FPSBA is often deemed inefficient in practical scenarios, as bidders are inclined to submit bids significantly lower than their actual valuation, resulting in suboptimal social welfare. In contrast, SPSBA, commonly called the Vickrey auction, requires the winner to pay the second-highest bid, encouraging truthful bidding by all players (Vickrey, 1961). It can be proven that bidding true valuations in SPSBA represents an NE. This auction evaluates agent performance in imperfect information games, where agents lack knowledge of other players' valuations.

## 2.3 SEQUENTIAL GAMES

**(7) Battle Royale**   Extended from the *Truel* (Kilgour, 1975) involving three players, the *Battle Royale* involves $N$ players shooting at each other. In the widely studied form (Kilgour & Brams, 1997), players have different probabilities of hitting the target, with the turn order set by increasing hit probabilities. The game allows for unlimited bullets and the tactical option of intentionally missing shots. The objective for each participant is to emerge as the sole survivor, with the game ending when only one player remains. While the NE has been identified for infinite sequential truels (Kilgour, 1977), the complexity of these equilibria escalates exponentially with an increased number of players.

**(8) Pirate Game**   This game is a multi-player version of the *Ultimatum Game* (Goodin, 1998; Stewart, 1999). Each player is assigned a "pirate rank", determining their action order. The game involves $N$ pirates discussing the division of $G$ golds they have discovered. The most senior pirate first proposes a distribution method. If the proposal is approved by at least half of the pirates, including the proposer, the game ends, and the gold is distributed as proposed. Otherwise, the most senior pirate is thrown overboard, and the next in rank assumes the proposer role until the game ends. Each pirate's objectives are prioritized as (1) survival, (2) maximizing their share of gold, and

(3) the opportunity to eliminate others from the game. Stewart (1999) identifies the optimal strategy, where the most senior pirate allocates one gold to each odd-ranked pirate and keeps the remainder.

## 3 GAMA-BENCH SCORING SCHEME

This section presents experiments conducted using the default settings for each game on the GPT-3.5 (0125) model. Utilizing this model as a case study, we illustrate our methodology for benchmarking an LLM with $\gamma$-Bench. The prompt and its design method can be found in §C in the appendix. Each game involves ten agents based on GPT-3.5, with the temperature parameter set to one. For simultaneous games, there will be twenty rounds. We run each game five times to enhance the reliability of our findings and mitigate the impact of variance. For clarity and conciseness, this section presents one of the five runs while §4.1 details quantitative results. Our findings of GPT-3.5's behaviors on $\gamma$-Bench include:

**Key Findings:**

- The model's decisions are mainly influenced by the outcomes of the preceding round rather than deriving from the reasoning of the optimal strategy.

- Although initially demonstrating suboptimal performance, the model can learn from historical data and enhance its performance over time. A larger fluctuation is observed in games that are difficult to optimize from historical data, such as the El Farol Bar game.

- The model demonstrates the ability to engage in spontaneous cooperation, leading to increased social welfare beyond mere self-interest, without the necessity for explicit communication. However, this phenomenon also results in low performance in Betraying Games.

- The model shows limitations in sequential games with more complicated rules.

- The aggregate score of the model on $\gamma$-Bench is 45.9.

### 3.1 COOPERATIVE GAMES

**(1) Guess 2/3 of the Average**  [TO PROMPT]  The vanilla setting for this game is $MIN = 0$, $MAX = 100$, and $R = \frac{2}{3}$. We show the choices made by all agents as well as the average and the winning numbers in Fig. 1(1). Key observations are: (1) In the first round, agents consistently select 50 (or close to 50), corresponding to the mean of a uniform distribution ranging from 0 to 100. This behavior suggests that the model fails to recognize that the winning number is $\frac{2}{3}$ of the average. (2) As rounds progress, the average number selected decreases noticeably, demonstrating that agents are capable of adapting based on historical outcomes. Since the optimal strategy is to choose the $MIN$, the score in this game is given by $S_1 = \frac{1}{NK} \sum_{ij} (C_{ij} - MIN)$, where $C_{ij}$ is the chosen number of player $i$ in round $j$. The model scores[2] 65.4 on this game.

**(2) El Farol Bar**  [TO PROMPT]  The vanilla setting for this game is $MIN = 0$, $MAX = 10$, $HOME = 5$, and $R = 60\%$. To explore the influence of incomplete information, we introduce two settings: *Explicit* indicates that everyone can see the results at the end of each round, while *Implicit* indicates that those staying at home cannot know what happened in the bar after the round ends. Fig. 1(2) illustrates the probability of agents deciding to go to the bar and the total number of players in the bar. We find that: (1) In the first round, there is an inclination among agents to visit the bar. Observations of overcrowding lead to a preference for staying home, resulting in fluctuations shown in both Fig. 1(2-1) and Fig. 1(2-2). In the Implicit setting, due to the lack of direct observations of the bar's occupancy, agents require additional rounds (Rounds 2 to 6) to discern the availability of space in the bar. (2) The probability of agents going to the bar gradually stabilizes, with the average probability in the Implicit setting being lower than in the Explicit setting. Since the optimal strategy is to choose the go with a probability of $R$, the raw score[3] in this game is given by $S_2 = \frac{1}{K} \sum_j |\frac{1}{N} \sum_i D_{ij} - R|$, where $D_{ij} = 1$ when player $i$ chose to go in round $j$ and $D_{ij} = 0$ when player $i$ chose to stay. The model scores 73.3 on this game.

---

[2]For clarity, we normalize raw scores to the range of $[0, 100]$, with higher values indicating a better performance. The method used for rescaling is detailed in §E of the appendix.

[3]For simplicity, we evaluate only the Implicit setting.

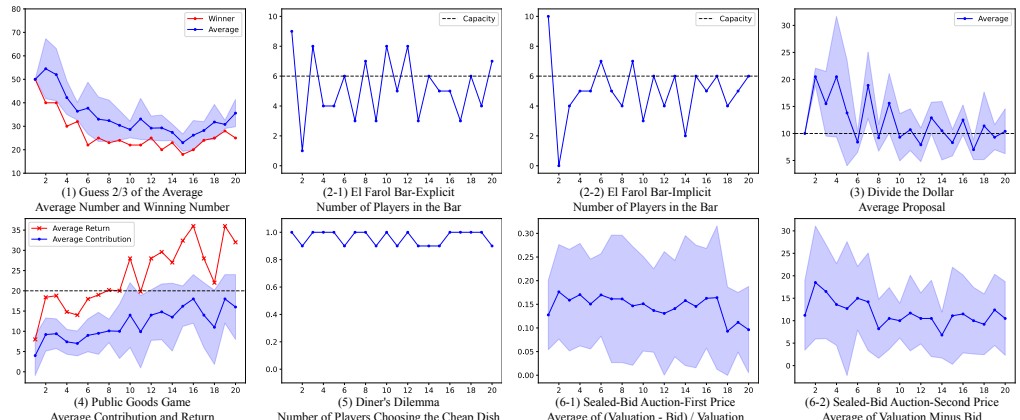

Figure 1: Performance of GPT-3.5 (0125) in Cooperative and Betraying games.

**(3) Divide the Dollar** [TO PROMPT] The vanilla setting for this game is $G = 100$. We plot the proposals by all agents and the sum of their proposals in Fig. 1(3). Our analysis reveals the following insights: (1) In the first round, agents' decisions align with the NE predictions of the game. However, after gaining golds, agents exhibit increased greed, proposing allocations that exceed the NE-prescribed amounts. Upon receiving nothing, they tend to propose a "safer" amount. The trend continues and causes fluctuations across subsequent rounds. (2) Despite these fluctuations, the average of proposed golds converges to approximately 100. Since the optimal strategy is to propose $G/N$, the raw score in this game is given by $S_3 = \frac{1}{K} \sum_j |\sum_i B_{ij} - G|$, where $B_{ij}$ is the proposed amount number of player $i$ in round $j$. The model scores $68.1$ on this game.

### 3.2 BETRAYING GAMES

**(4) Public Goods Game** [TO PROMPT] The vanilla setting for this game is $R = 2$. Each player has $T = 20$ to contribute in each round. Fig. 1(4) shows the contributed tokens by each agent and their corresponding gains per round. The observations reveal the following: (1) Despite an investment return of $-80\%$, agents display a pattern of alternating between free-riding and contributing all their tokens. (2) As the rounds progress, there is an evident increase in the number of tokens contributed to the public pot, leading to an overall enhancement in social welfare gains. These findings suggest that the LLM exhibits cooperative behavior, prioritizing collective benefits over individual self-interest. Since we expect the model to infer the optimal strategy, *i.e.*, contributing zero tokens, the raw score in this game is given by $S_4 = \frac{1}{NK} \sum_{ij} C_{ij}$, where $C_{ij}$ is the proposed contribution amount of player $i$ in round $j$. The model scores $41.2$ on this game.

**(5) Diner's Dilemma** [TO PROMPT] The vanilla setting for this game is $P_h = 20$, $P_l = 10$, $U_h = 20$, $U_l = 15$. We show the probability of agents choosing the costly dish, their resulting utilities, and the average bill in Fig. 1(5). Analysis of the figure reveals the following insights: (1) Contrary to the NE predictions for this game, agents predominantly prefer the cheap dish, which maximizes total social welfare. (2) Remarkably, a deviation from cooperative behavior is observed wherein one agent consistently chooses to betray others, thereby securing a higher utility. This pattern of betrayal by this agent persists across subsequent rounds. Since we expect the model to infer the the optimal strategy, *i.e.*, choosing the costly dish, the raw score in this game is given by $S_5 = \frac{1}{NK} \sum_{ij} D_{ij}$, where $D_{ij} = 1$ when player $i$ chose the cheap dish in round $j$ and $D_{ij} = 0$ when player $i$ chose the costly dish. The model scores $4.0$ on this game.

**(6) Sealed-Bid Auction** [TO PROMPT] For the vanilla setting in this game, we randomly assign valuations to each agent in each round, ranging from 0 to 200. We fix the seed for random number generation to ensure fair comparisons across various settings and models. We evaluate LLMs' performance under both *First-Price* and *Second-Price* settings. Fig. 1(6) depicts the subtraction between valuations and bids and bid amounts of each agent. Our key findings include: (1) As introduced in §2.2, we note that agents generally submit bids that are lower than their valuations in the First-Price auction, a tendency indicated by the positive discrepancies between valuations and bids

depicted in Fig. 1(6-1). (2) Though the NE suggests that everyone bids the amount of their valuation in the Second-Price setting, we find a propensity for bidding below valuation levels, as demonstrated in Fig. 1(6-2). Since the optimal strategy is to bid the prices lower than their true valuations,[4] the raw score in this game is given by $S_6 = \frac{1}{NK} \sum_{ij} \frac{v_{ij} - b_{ij}}{v_{ij}}$, where $v_{ij}$ and $b_{ij}$ are player $i$'s valuation and bid in round $j$, respectively. The model scores 14.6 on this game.

## 3.3 SEQUENTIAL GAMES

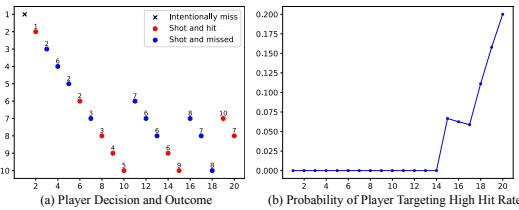

(a) Player Decision and Outcome  (b) Probability of Player Targeting High Hit Rate

Figure 2: GPT-3.5 (0125)'s performance in "Battle Royale." (a): Agents' actions and outcomes of each round. For example, in round 11, player 6 shot at player 7 but missed.

**(7) Battle Royale** [TO PROMPT] For the vanilla setting in this game, we assign varied hit rates to each agent, spanning from 35% to 80% in increments of 5%. This setting covers a broad spectrum of hit rates, avoiding extremes of 0% or 100%. Fig. 2 illustrates the actions and outcomes of each round, along with the tally of participants remaining. Our observations reveal: (1) Unlike our expectations, agents rarely target the player with the highest hit rate. (2) Agents neglect to utilize the strategy of "intentionally missing." For example, in round 19, with players 7, 8, and 10 remaining, it was player 7's turn to act. The optimal strategy for player 7 would have been to intentionally miss the shot, thereby coaxing player 8 into eliminating player 10, enabling player 7 to target player 8 in the following round for a potential victory. Instead, player 7 opted to target player 10, resulting in player 8 firing upon itself. For simplicity, we evaluate whether agents target the player with the highest hit rate (excluding themselves). Therefore, the raw score in this game is given by $S_7 = \frac{1}{Nk} \sum_{ij} I_{ij}$, where $k$ represents the number of rounds played and $I_{ij} = 1$ if player $i$ targets the player with the highest hit rate in round $j$, and $I_{ij} = 0$ otherwise. The model scores 20.0 on this game.

Table 2: Performance of GPT-3.5 (0125) in the "Pirate Game." Each row shows the proposed gold distribution in the specific round and whether each pirate accepts ("✓") or rejects ("✗") the proposal. $S_{8P}$ shows the score of the proposer while $S_{8V}$ shows the score of all voters.

| Pirate Rank | 1 | 2 | 3 | 4 | 5 | 6 | 7 | 8 | 9 | 10 | $S_{8P}$ | $S_{8V}$ |
|---|---|---|---|---|---|---|---|---|---|---|---|---|
| **Round 1** | 100✓ | 0✗ | 0✗ | 0✗ | 0✗ | 0✗ | 0✗ | 0✗ | 0✗ | 0✗ | 8 | 1.00 |
| **Round 2** | - | 99✓ | 0✗ | 1✓ | 0✓ | 0✗ | 0✗ | 0✗ | 0✗ | 0✓ | 6 | 0.75 |
| **Round 3** | - | - | 50✓ | 1✓ | 1✓ | 1✓ | 1✓ | 1✓ | 1✓ | 44✓ | 94 | 0.57 |

**(8) Pirate Game** [TO PROMPT] The vanilla setting for this game is $G = 100$. As introduced in §2.3, the optimal strategy for the first proposer is to allocate 96 golds to itself and one gold each to the third, fifth, seventh, and ninth pirates. Stewart (1999) has elucidated the optimal strategy for voters: (1) accept if allocated two or more golds; (2) reject if no golds are allocated; (3) accept if one gold is allocated and it shares the same parity as the proposer, otherwise, reject. Table 2 presents a sample game's proposals and voting results. The key conclusion is that agents fail to propose optimal proposals and frequently cast incorrect votes, suggesting that the LLM demonstrates suboptimal performance in this game. Two aspects are considered to comprehensively evaluate a model's performance: (1) whether proposers give a reasonable proposal and (2) whether voters act correctly towards a given proposal. For (1), we calculate the $L_1$ norm between the given proposal and the optimal strategy, defined as $S_{8P} = \frac{1}{k} \sum_j \|P_j - O_j\|_1$, where $P_j$ represents the model's proposal and $O_j$ denotes the optimal proposal in round $j$, with the game ending at round $k$. For (2), we calculate the accuracy of choosing the right action elucidated above, which is: $S_{8V} = \frac{2}{k(2N-k-1)} \sum_{ij} I_{ij}$, where $I_{ij} = 1$ if player $i$ votes correctly in round $j$ and $I_{ij} = 0$ otherwise, excluding the proposer from the calculation. The model scores 80.6 on this game.

---

[4]We evaluate only the First-Price setting according to the definition of Betraying Games.

## 4 BEYOND DEFAULT SETTINGS

This section explores deeper into several following Research Questions (RQs). **RQ1 Robustness**: Is there a significant variance in multiple runs? Is the performance sensitive to different temperatures and prompt templates? **RQ2 Reasoning Strategies**: Are strategies to enhance reasoning skills applicable to game scenarios? This includes implementing Chain-of-Thought (CoT) (Kojima et al., 2022; Wei et al., 2022) reasoning and assigning unique personas to LLMs. **RQ3 Generalizability**: How does LLM performance vary with different game settings? Do LLMs remember answers learned during the training phase? **RQ4 Leaderboard**: How do various LLMs perform on $\gamma$-Bench? Unless otherwise specified, we apply the vanilla settings described in §3.

### 4.1 RQ1: ROBUSTNESS

This RQ examines the stability of LLMs' responses, assessing the impact of three critical factors on model performance: (1) randomness introduced by the model's sampling strategy, (2) the temperature parameter setting, and (3) the prompt used for game instruction.

**Multiple Runs**  Firstly, we run all games five times under the same settings. Fig. 4 illustrates the average performance across tests, while Table 4 lists the corresponding scores. The analysis reveals that, except for the two sequential games and the "Public Goods Game," the model demonstrates a consistent performance, as evidenced by the low variance in scores for each game.

**Temperatures**  As discussed in our literature review in §B, prior research incorporates varying temperature parameters from $0$ to $1$ yet omits to explore their impacts. This study conducts experiments across games employing a range of temperatures $\{0.0, 0.2, 0.4, 0.6, 0.8, 1.0\}$ under vanilla settings. The results, both visual and quantitative, are documented in Fig. 5 and Table 5, respectively. The small overall variance of $3.4$ indicates that, for the majority of games, temperature adjustments yield negligible effects. A notable exception is observed in "Guess 2/3 of the Average," where increased temperatures correlate with enhanced scores ($48.0$ to $65.4$), contrasting starkly with the near-random performance at zero temperature.

**Prompt Templates**  We also investigate the impact of prompt phrasing on model performance. We leverage GPT-4 to rewrite our default prompt templates, generating four additional versions. We perform a manual checking process on the generated versions to ensure GPT-4's adherence to game rules without altering critical data. The prompt templates can be found in §D. We plot the results of using these templates in Fig. 6 and record the quantitative scores in Table 6. Notably, we find that prompt wording can significantly affect performance, as shown by the high variances in the "Public Goods Game" ($11.5$), "Diner's Dilemma" ($23.7$), and "Pirate Game" ($14.7$).

> **Answer to RQ1:** GPT-3.5 exhibits consistency in multiple runs and shows robustness against different temperature settings. However, inappropriate prompt designs resulting from potential misinformation during rephrasing can significantly impair performance.

### 4.2 RQ2: REASONING STRATEGIES

This RQ focuses on improving the model's performance through prompt instructions. We investigate two strategies: Chain-of-Thought (CoT) prompting (Kojima et al., 2022) and persona assignment (Kong et al., 2024). We show the visualized and quantitative results in Fig. 7 and Table 7.

**CoT**  According to Kojima et al. (2022), introducing a preliminary phrase, "Let's think step by step," encourages the model to sequentially analyze and explain its reasoning before presenting its conclusion. This approach has proven beneficial in specific scenarios, such as games (1), (3), (4), and (5), improving the overall score from $45.9$ to $57.9$, by $12.0$. In the "(3) Divide the Dollar" game, incorporating CoT reduces the model's propensity to suggest disproportionately large allocations, increasing the score by $15.3$. Similarly, in the "(4) Public Goods Game" and "(5) Diner's Dilemma," CoT prompts the model to recognize being a free-rider as the optimal strategy, increasing the scores by $14.9$ and $78.5$, respectively.

**Persona** Studies (Kong et al., 2024; Huang et al., 2024b) have demonstrated that assigning roles to models influences performance across various downstream tasks. Inspired by this discovery, our study initiates with a prompt that specifies the model's role, such as "You are [ROLE]," where the role could be a cooperative and collaborative assistant, a selfish and greedy assistant, or a mathematician. Our findings reveal that assigning the "cooperative" role enhances model performance in games (1), (2), and (3), notably outperforming the CoT method in the "(3) El Farol Bar" game. Conversely, the "selfish" role markedly diminishes performance almost all the games, with the only exception of the "(7) Battle Royale" game. The "mathematician" role improves the model's overall score by 0.6, which is small and does not surpass the CoT method's effectiveness.

> **Answer to RQ2:** It is possible to improve GPT-3.5 through simple prompt instructions. Among the methods we explore, the CoT prompting performs the best, achieving a performance close to GPT-4 (57.9 vs. 62.4).

### 4.3 RQ3: GENERALIZABILITY

Considering the extensive exploration of games in domains such as mathematics, economics, and computer science, it is probable that the vanilla settings of these games are included within the training datasets of LLMs. To ascertain the presence of data contamination in our chosen games, we subjected them to various settings. The specifics of the parameters selected for each game are detailed in Table 8, and the experimental outcomes are visually represented in Fig. 8. Our findings indicate variability in model generalizability across different games. Specifically, in games (1), (3), (5), (6), and (8), the model demonstrated correct performance under diverse settings. In the "(3) Divide the Dollar" game, the model's performance improved with an increase in total golds ($G$), suggesting that higher allocations of golds satisfy the demands of all players. Conversely, the model exhibited low generalizability in games (2) and (4). An analysis of the game "(2) El Farol Bar" reveals a consistent decision-making pattern by the model, opting to participate with approximately a $50\%$ probability regardless of varying bar capacities ($R$), indicating that the model is acting randomly. Similarly, in the "(4) Public Goods Game," the model consistently contributes similar amounts, even when the return rate is nil, indicating a lack of understanding of the game rules. A possible reason for this poor performance is the model's inability to adjust its performance incrementally based on historical data.

Nagel (1995) conducted experiments with 15 to 18 human subjects participating in the "(1) Guess 2/3 of the Average" game, using ratios of $\frac{1}{2}$, $\frac{2}{3}$, and $\frac{4}{3}$ . The average numbers were 27.05, 36.73, and 60.12 for each ratio, respectively. In a similar vein, Rubinstein (2007) explored the $\frac{2}{3}$ ratio on a larger population involving 2,423 subjects, yielding a comparable mean of 36.2, aligning with the finding in Nagel (1995). The model produces average numbers of 34.59, 34.59, and 74.92 for the same ratios, indicating its predictions are more aligned with human behavior than the game's NE.

> **Answer to RQ3:** GPT-3.5 demonstrates variable performance across different game settings, exhibiting notably lower efficacy in "(2) El Farol Bar" and "(4) Public Goods Game." It is noteworthy that, $\gamma$-Bench provides a test bed to evaluate the ability of LLMs in complex reasoning scenarios. As model's ability improves (*e.g.*, achieving more than 90 on $\gamma$-Bench), we can increase the difficulty by varying game settings.

### 4.4 RQ4: LEADERBOARD

This RQ investigates the variance in decision-making capabilities among different LLMs, using $\gamma$-Bench. We first focus on closed-source models, including OpenAI's GPT-3.5 (0613, 1106, and 0125), GPT-4 (Turbo-0125, 4o-0806), and Google's Gemini Pro (1.0, 1.5). The results are organized in Table 1a, with model performance visualized in Fig. 9 in the appendix. Gemini-1.5-Pro scores 69.8, markedly surpassing other models, particularly in games (1), (4), and (5). GPT-4o follows closely behind Gemini-Pro, achieving 66.75. GPT-4's lowered performance in the "(2) El Farol Bar" game (23.0) and the "(5) Diner's Dilemma" game (0.9) stems from its conservative strategies favoring staying at home and spending less money. Similarly, the "(6) Sealed-Bid Auction" (24.2) is attributed to a strategy of not risking bidding high or low. The risk-averse preference also explains the relatively good score on the "(4) Public Goods Game," where the GPT-4 does not take the risk to invest. Furthermore, an evaluation of three GPT-3.5 updates shows similar performance.

Next, we focus on open-source models, whose performance is detailed in Table 1b and visualized in Fig. 10. The top-two open-source model, LLaMA-3.1-70B and Mixtral-8x22B, closely follows Gemini-1.5-Pro with a score of 65.9 and 62.4, surpassing GPT-4. Most open-source models, including Qwen-2, LLaMA-3.1-405B, and LLaMA-3.1-8B, outperform GPT-3.5 and Gemini-1.0-Pro. Mixtral-8x7B exhibits the lowest performance, likely due to its smaller size and weaker reasoning capabilities. Interestingly, LLaMA-3.1-405B underperforms compared to its smaller counterpart, the 70B version, which we attribute to its overly conservative strategy in the "(2) El Farol Bar" game, a challenge similar to the one faced by GPT-4.

**Answer to RQ4:** Currently, Gemini-1.5-Pro outperforms all other models evaluated in this study. LLaMA-3.1-70B performs closely, being in the second place.

## 5 RELATED WORK

Evaluating LLMs through game theory models has become a popular research direction. An overview on recent studies is summarized in Table 3. We find: (1) Many studies examine the PSNE on two-player, single-round settings, focusing on the *Prisoner's Dilemma* and the *Ultimatum Game*. (2) Varying temperatures are employed without discussing the impact on LLMs' performance.

### 5.1 SPECIFIC GAMES

Researchers have explored diverse game scenarios. Using the complex and deceptive environments of *Avalon* game as a test bed, recent work focuses on long-horizon multi-party dialogues (Stepputtis et al., 2023), social behaviors (Lan et al., 2024), social intelligence (Liu et al., 2024), and recursive contemplation (Wang et al., 2023) for identifying deceptive information. Other papers have investigated communication games like *Werewolf*, with a focus on tuning-free frameworks (Xu et al., 2023) and reinforcement learning-powered approaches (Xu et al., 2024b). O'Gara (2023) found that advanced LLMs exhibit deception and lie detection capabilities in the text-based game, *Hoodwinked*. Meanwhile, Liang et al. (2023) evaluated LLMs' intelligence and strategic communication skills in the word guessing game, *Who Is Spy?* In the game of *Water Allocation Challenge*, Mao et al. (2025) constructed a scenario highlighting unequal competition for limited resources.

### 5.2 GAME BENCHMARKS

Another line of studies collects games to build more comprehensive benchmarks to assess the artificial general intelligence of LLMs. Tsai et al. (2023) found that while LLMs perform competitively in text games, they struggle with world modeling and goal inference. GameEval (Qiao et al., 2023) introduced three goal-driven conversational games (*Ask-Guess*, *SpyFall*, and *TofuKingdom*) to assess the problem-solving capabilities of LLMs in cooperative and adversarial settings. MAgIC (Xu et al., 2024a) proposed the probabilistic graphical modeling method for evaluating LLMs in multi-agent game settings. LLM-Co (Agashe et al., 2023) assesses LLMs in multi-agent coordination scenarios, showcasing their capabilities in partner intention inference and proactive assistance. SmartPlay (Wu et al., 2024) evaluated LLMs as agents across six games, emphasizing reasoning, planning, and learning capabilities. Abdelnabi et al. (2024) designed negotiation games involving six parties with distinct objectives to evaluate LLMs' ability to reach agreement.

## 6 CONCLUSION

This paper presents $\gamma$-Bench, a benchmark designed to assess LLMs' Gaming Ability in Multi-Agent environments. $\gamma$-Bench incorporates eight classic game theory scenarios, emphasizing multiplayer interactions across multiple rounds and actions. Our findings reveal that GPT-3.5 (0125) demonstrates a limited decision-making ability on $\gamma$-Bench, yet it can improve itself by learning from the historical results. Leveraging the carefully designed scoring scheme, we observe that GPT-3.5 (0125) exhibits commendable robustness across various temperatures and prompts. It is noteworthy that strategies such as CoT prove effective in this context. Nevertheless, its capability to generalize across various game settings remains restricted. Finally, Gemini-1.5-Pro outperforms all tested models, achieving the highest ranking on the $\gamma$-Bench leaderboard, with the open-source LLaMA-3.1-70B following closely behind.

## ACKNOWLEDGMENTS

The paper is supported by the Research Grants Council of the Hong Kong Special Administrative Region, China (No. CUHK 14206921 of the General Research Fund).

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

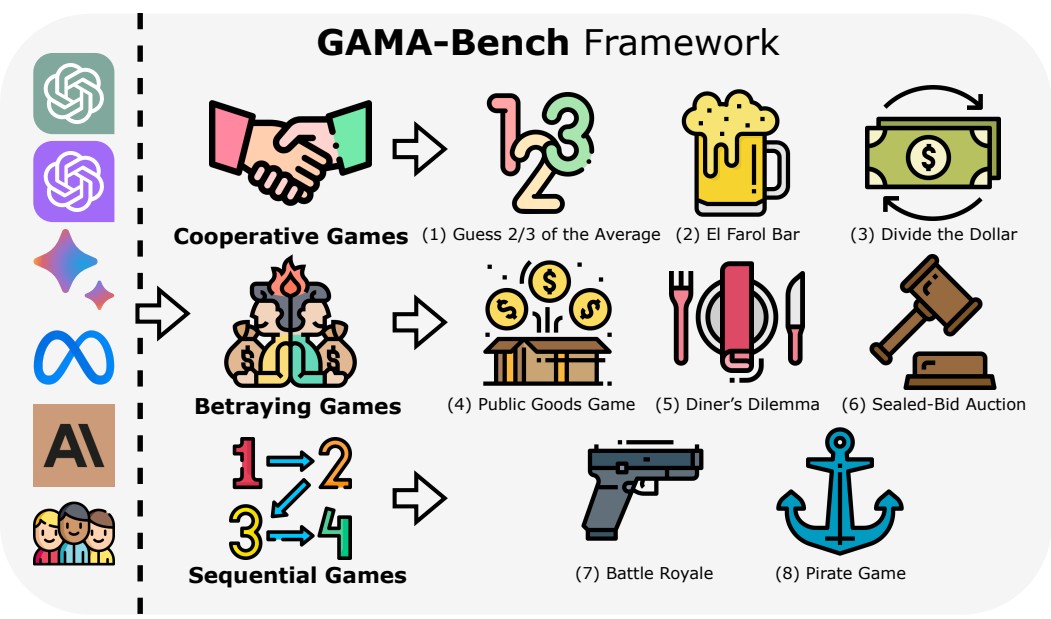

Figure 3: $\gamma$-Bench enables multiple LLMs and humans to engage in multi-round games. The framework comprises three categories of games, each targeting different LLM abilities, and includes eight classic games from *Game Theory*.

## CONTENTS

# A   MORE INFORMATION ON GAME THEORY

## A.1   FORMULATION

Game theory involves analyzing mathematical models of strategic interactions among rational agents (Myerson, 2013). A game can be modeled using these key elements:

1. Players, denoted as $\mathcal{P} = \{1, 2, \cdots, N\}$: A set of $N$ participants.
2. Actions, represented as $\mathcal{A} = \{\mathcal{A}_i\}$: $N$ sets of actions available to each player. For instance, $\mathcal{A} = \{\mathcal{A}_1 = \{C, D\}, \mathcal{A}_2 = \{D, F\}, \cdots, \mathcal{A}_N = \{C, F\}\}$
3. Utility functions, denoted as $\mathcal{U} = \{\mathcal{U}_i\colon \times_{j=1}^N \mathcal{A}_j \mapsto \mathbb{R}\}$: A set of $N$ functions that quantify each player's preferences over all possible outcomes.
4. Information, represented as $\mathcal{I} = \{\mathcal{I}_i\}$: $N$ sets of information available to each player, including other players' action sets, utility functions, historical actions, and other beliefs.
5. Order, indicated by $\mathcal{O} = \mathcal{O}_1, \mathcal{O}_2, \cdots, \mathcal{O}_k$: A sequence of $k$ sets specifying the $k$ steps to take actions. For example, $\mathcal{O} = \mathcal{P}$ implies that all players take actions simultaneously.

In this study, *Multi-Player* games are defined as those with $|\mathcal{P}| > 2$ since game theory models have at least two players. Similarly, *Multi-Action* games are those where $\forall_{i \in \mathcal{P}} |\mathcal{A}_i| > 2$. Meanwhile, *Multi-Round* games involve the same set of players repeatedly engaging in the game, with a record of all previous actions being maintained. *Simultaneous* games satisfy that $k = 1$, whereas *Sequential* games have $k > 1$, indicating players make decisions in a specific order. Games of *Perfect Information* are characterized by the condition $\forall_{i,j \in \mathcal{P} | i \neq j} \mathcal{I}_i = \mathcal{I}_j$. Since every player can see their own action, the above condition indicates that all players are visible to the complete information set in the game. Conversely, games not meeting this criterion are classified as *Imperfect Information* games, where players have limited knowledge of others' actions.

## A.2   NASH EQUILIBRIUM

Studying game theory models often involves analyzing their Nash Equilibria (NE) (Nash, 1950). An NE is a specific set of strategies where no one has anything to gain by changing only one's own strategy. This implies that given one player's choice, the strategies of others are constrained to a specific set, which in turn limits the original player's choice to the initial one. When each player's strategy contains only one action, the equilibrium is identified as a *Pure Strategy Nash Equilibrium* (PSNE) (Nash, 1950). However, in certain games, such as rock-paper-scissors, an NE exists only when players employ a probabilistic approach to their actions. This type of equilibrium is known as a *Mixed Strategy Nash Equilibrium* (MSNE) (Nash, 1951), with PSNE being a subset of MSNE where probabilities are concentrated on a single action. According to Thm. A.1 shown below, we can analyze the NE of each game and evaluate whether LLMs' choices align with the NE.

**Theorem A.1 (Nash's Existence Theorem)** *Every game with a finite number of players in which each player can choose from a finite number of actions has at least one mixed strategy Nash equilibrium, in which each player's action is determined by a probability distribution.*

## A.3   HUMAN BEHAVIORS

The attainment of NE presupposes participants as *Homo Economicus*, who are consistently rational and narrowly self-interested, aiming at maximizing self goals (Persky, 1995). However, human decision-making often deviates from this ideal. Empirical studies reveal that human choices frequently diverge from what the NE predicts (Nagel, 1995). This deviation is attributed to the complex nature of human decision-making, which involves not only rational analysis but also personal values, preferences, beliefs, and emotions. By comparing human decision patterns documented in prior studies, together with the NE, we can ascertain whether LLMs exhibit tendencies more akin to homo economicus or actual human decision-makers, thus shedding light on their alignment with human-like or purely rational decision-making processes.

# B LITERATURE REVIEW: EVALUATING LLMs WITH GAME THEORY

Evaluating LLMs through game theory models has become a popular research direction. An overview on recent studies is summarized in Table 3. From our analysis, several key observations emerge: (1) The majority of these studies are concentrated on two-player settings. (2) There is a predominant focus on two-action games; notably, half of the studies examine the *Prisoner's Dilemma* and the *Ultimatum Game* (the *Dictator Game* is one of the variants of the *Ultimatum Game*). (3) A notable gap in the literature is the lack of the comparative studies between LLMs' decision-making across multiple rounds and the action probability distributions predicted by the MSNE. (4) The studies exhibit variability in the temperatures used, which precludes definitive conclusions regarding their impact on LLMs' performance.

Table 3: A Comparison of existing studies that evaluate LLMs using game theory models. **T** denotes the temperature employed in each experiment. **MP** refers to a multi-player setting, whereas **MR** indicates multi-round interactions. **Role** specifies whether a specific role is assigned to the LLMs.

| Paper | Models | T | MP | MR | Role | CoT | Games |
|---|---|---|---|---|---|---|---|
| Horton (2023) | text-davinci-003 | - | ✗ | ✗ | ✗ | ✗ | Dictator Game |
| Guo (2023) | gpt-4-1106-preview | 1 | ✗ | ✓ | ✓ | ✓ | Ultimatum Game, Prisoner's Dilemma |
| Phelps & Russell (2023) | gpt-3.5-turbo | 0.2 | ✗ | ✓ | ✓ | ✗ | Prisoner's Dilemma |
| Akata et al. (2023) | text-davinci-003, gpt-3.5-turbo, gpt-4 | 0 | ✗ | ✓ | ✗ | ✗ | Prisoner's Dilemma, Battle of the Sexes |
| Aher et al. (2023) | text-ada-001, text-babbage-001, text-curie-001, text-davinci-001, text-davinci-002, text-davinci-003, gpt-3.5-turbo, gpt-4 | 1 | ✗ | ✗ | ✓ | ✗ | Ultimatum Game |
| Capraro et al. (2023) | ChatGPT-4, Bard, Bing Chat | - | ✗ | ✗ | ✗ | ✓ | Dictator Game (Three Variants) |
| Brookins & DeBacker (2024) | gpt-3.5-turbo | 1 | ✗ | ✗ | ✗ | ✗ | Dictator Game, Prisoner's Dilemma |
| Li et al. (2023) | gpt-3.5-turbo-0613, gpt-4-0613, claude-2.0, chat-bison-001 | - | ✓ | ✓ | ✗ | ✗ | Public Goods Game |
| Heydari & Lorè (2023) | gpt-3.5-turbo-16k, gpt-4, llama-2 | 0.8 | ✗ | ✗ | ✓ | ✓ | Prisoner's Dilemma, Stag Hunt, Snowdrift, Prisoner's Delight |
| Guo et al. (2024) | gpt-3.5, gpt-4 | - | ✗ | ✓ | ✗ | ✓ | Leduc Hold'em |
| Chen et al. (2024) | gpt-3.5-turbo-0613, gpt-4-0613, claude-instant-1.2, claude-2.0, chat-bison-001 | 0.7 | ✓ | ✓ | ✓ | ✓ | English Auction |
| Xu et al. (2024a) | gpt-3.5-turbo, gpt-4, llama-2-70b, claude-2.0, palm-2 | - | ✓ | ✓ | ✗ | ✓ | Cost Sharing, Prisoner's Dilemma, Public Goods Game |
| Fan et al. (2024) | text-davinci-003, gpt-3.5-turbo, gpt-4 | 0.7 | ✗ | ✓ | ✗ | ✗ | Dictator Game, Rock-Paper-Scissors, Ring-Network Game |
| Zhang et al. (2024) | gpt-4 | 0.7 | ✓ | ✓ | ✓ | ✓ | Guess 0.8 of the Average Survival Auction Game |
| Duan et al. (2024) | gpt-3.5-turbo, gpt-4, llama-2-70b, codellama-34b, mistral-7b-orca | 0.2 | ✓ | ✓ | ✗ | ✓ | Ten Games[a] |
| Xie et al. (2024) | text-davinci-003, gpt-3.5-turbo-instruct, gpt-3.5-turbo-0613, gpt-4, llama-2-(7/13/70)b, vicuna-(7/13/33)b-v1.3 | - | ✗ | ✓ | ✓ | ✓ | Seven Games[b] |
| **This Study** | gpt-3.5-turbo, gpt-4, gemini-pro | 0∼1 | ✓ | ✓ | ✓ | ✓ | Eight Games[c] |

[a] Tic-Tac-Toe, Connect-4, Kuhn Poker, Breakthrough, Liar's Dice, Blind Auction, Negotiation, Nim, Pig, Iterated Prisoner's Dilemma.
[b] Trust Game, Minimum Acceptable Probabilities Trust Game, Repeated Trust Game, Dictator Game, Risky Dictator Game, Lottery People Game, Lottery Gamble Game.
[c] Guess 2/3 of the Average, El Farol Bar, Divide the Dollar, Public Goods Game, Diner's Dilemma, Sealed-Bid Auction, Battle Royale, Pirate Game.

[BACK TO INTRODUCTION]

[BACK TO RELATED WORK]

## C  DETAILS ABOUT PROMPTS

### C.1  DESIGN METHODOLOGY

We adopt a cohesive approach to ensure the prompt design is systematic and not arbitrary. Game descriptions are gathered from verified sources, including academic papers referenced in §2 and Wikipedia entries. Using these descriptions, we instruct GPT-4 to generate prompts to guide LLMs in engaging in the specified games. These prompts are structured to encompass four essential elements: the rules of the game, objectives for the players, a template for announcing game outcomes (for displaying historical results), and instructions for formatting responses in JSON. A manual checking process is conducted to ascertain that GPT-4's comprehension of the game descriptions is correct. The prompts are detailed in the rest part of this section.

### C.2  COOPERATIVE GAMES

| **Guess 2/3 of the Average** | |
|---|---|
| SYSTEM | You are participating in a game played by $N$ players over $K$ rounds. |
| | Game Rules: |
| | 1. Each player selects an integer number between $MIN$ and $MAX$, inclusive. |
| | 2. After all selections are made, the average of all chosen numbers is calculated. |
| | 3. The target number is $R$ of this average. |
| | 4. The winner is the player(s) who selected a number closest to the target number. |
| | $\cdots$ |
| USER | Game Results for Round $I$: |
| | Average Number Chosen: $M_I$ |
| | Target Number ($R$ of Average): $T_I$ |
| | Winning Number: $W_I$ |
| | You chose: |
| ASSISTANT | {"chosen_number": "$C_{IJ}$"} |
| USER | [Congratulation you won]/[Unfortunately you lost]. |
| | $\cdots$ |
| USER | Now round $I$ starts. |
| | Your goal is to choose a number that you believe will be closest to $R$ of the average of all numbers chosen by players, including your selection. |
| | Please provide your chosen number in the following JSON format: |
| | {"chosen_number": "integer_between_$MIN$_and_$MAX$"}. |

[BACK TO GAME DESCRIPTION]

[BACK TO VANILLA EXPERIMENT]

**El Farol Bar**

| | |
|---|---|
| SYSTEM | You are participating in a game played by $N$ players over $K$ rounds. |
| | Game Rules: |
| | 1. Every round, you and the other players decide independently whether to go to a bar. |
| | 2. If equal to or less than $R$ of the players go to the bar, everyone who goes has more fun than staying home, receiving a utility of $MAX$. |
| | 3. If more than $R$ of the players go to the bar, everyone who goes has less fun than staying home, receiving a utility of $MIN$. |
| | 4. Everyone who stays home receives a utility of $HOME$. |
| | $\cdots$ |
| USER | Game Results for Round $I$: |
| | (Only for Explicit) $G_I$ players went to the bar, while $S_I$ players stayed home. |
| | $G_I/N$, which is [more]/[equal to or less] than $R$ of the players went to the bar. |
| | It was [less]/[more] fun to go to the bar this round. |
| | You chose: |
| ASSISTANT | {"decision": "$D_{IJ}$"} |
| USER | You gained $G_{IJ}$. |
| | $\cdots$ |
| USER | Now round $I$ starts. |
| | Your goal is to maximize your fun. Choose to go to the bar when you predict fewer than $R$ of the players will go, and choose to stay home otherwise. |
| | Please provide your decision in the following JSON format: |
| | {"decision": "go_or_stay"}. |

**Divide the Dollar**

| | |
|---|---|
| SYSTEM | You are participating in a game played by $N$ players over $K$ rounds. |
| | Game Rules: |
| | 1. You are dividing $G$ golds. Each player independently proposes a bid. |
| | 2. If the sum of all bids does not exceed $G$, each player receives their bid amount. |
| | 3. If the sum exceeds $G$, all players receive nothing. |
| | $\cdots$ |
| USER | Game Results for Round $I$: |
| | Your bid amount was: |
| ASSISTANT | {"bid_amount": "$B_{IJ}$"} |
| USER | The sum of all bids was $S_I$. |
| | The sum [does not exceed]/[exceeds] $G$. |
| | You received [$B_{IJ}$]/[0] golds. |
| | $\cdots$ |
| USER | Now round $I$ starts. |
| | Your goal is to maximize your individual gain without causing the total sum of bids to exceed $G$ golds. |
| | Please provide your bid amount in the following JSON format: |
| | {"bid_amount": "integer_between_0_and_G"}. |

## C.3 BETRAYING GAMES

---

**Public Goods Game**

| | |
|---|---|
| SYSTEM | You are participating in a game played by $N$ players over $K$ rounds. |
| | Game Rules: |
| | 1. In each round, you, as a player, must decide how many of your private tokens you wish to contribute secretly to the public pot. |
| | 2. The total tokens in this pot will be multiplied by the factor $R$ to create the "public good" payoff. |
| | 3. This payoff will then be evenly divided among all players, regardless of their individual contribution. |
| | 4. Any tokens you do not contribute will be retained in your private collection. |
| | $\cdots$ |
| USER | Game Results for Round $I$: |
| | Contributed tokens of each player: $C_{I1}, C_{I2}, \cdots, C_{IN}$ |
| | You contributed: |
| ASSISTANT | {"tokens_contributed": "$C_{IJ}$"} |
| USER | Tokens in the public pot: $S_I$ |
| | Your gain: $g_{IJ}$ |
| | Your tokens after round $I$: $T_{IJ}$ |
| | Tokens of each player after round $I$: $T_{I1}, T_{I2}, \cdots, T_{IN}$ |
| | $\cdots$ |
| USER | Now round $I$ starts. |
| | Your goal is to maximize your total token count by the end of the game. Currently you have $T_{I-1J}$ tokens. You need to decide the number of tokens to be contributed to the public pot. |
| | Please provide the number of tokens in the following JSON format: |
| | {"tokens_contributed": "integer_between_0_and_$T_{IJ}$"} |

---

**Diner's Dilemma**

| | |
|---|---|
| SYSTEM | You are participating in a game played by $N$ players over $K$ rounds. |
| | Game Rules: |
| | 1. Each player must choose to order either a costly dish or a cheap dish. |
| | 2. The price of the costly dish is $P_h$. The price of the cheap dish is $P_l$. |
| | 3. The costly dish brings you a utility of $U_h$. The cheap dish brings you a utility of $U_l$. |
| | 4. The costly dish is tastier than the cheap dish, but not sufficiently to justify its price when dining alone. |
| | 5. At the end of each round, the total cost of all dishes ordered is split equally among all players. |
| | $\cdots$ |
| USER | Game Results for Round $I$: |
| | $N_h$ people chose the costly dish, while $N_l$ chose the cheap dish. |
| | The total cost is $S_I$. You need to pay $C_I$. |
| | You chose: |
| ASSISTANT | {"chosen_dish": "$D_{IJ}$"} |
| USER | Your utility is $u_{IJ}$. |
| | $\cdots$ |
| USER | Now round $I$ starts. |
| | Your goal is to maximize your overall satisfaction, balancing the quality of the dish and the cost shared. |
| | Please provide your chosen dish in the following JSON format: |
| | {"chosen_dish": "costly_or_cheap"} |

---

| **Sealed-Bid Auction** |
|---|

| SYSTEM | You are participating in a game played by $N$ players over $K$ rounds. |
|---|---|
| | Game Rules: |
| | 1. Each player has a private valuation for the item in each round. |
| | 2. Without knowing the bids and valuations of other players, each player submits a written bid for the item. |
| | 3. The highest bidder wins the item and pays the price of the [highest]/[second highest] bid. |
| | 4. If you win, your utility for that round is your valuation minus the price paid. If you lose, your utility is zero. |
| | $\cdots$ |
| USER | Game Results for Round $I$: |
| | Your valuation for this round's item was $v_{IJ}$. |
| | Your bid was: |
| ASSISTANT | {"bid": "$b_{IJ}$"} |
| USER | The winning bid was: $W_I$. |
| | The price paid was: $P_I$. |
| | You [won]/[lost]. Your utility is [$u_{IJ}$]/[0]. |
| | $\cdots$ |
| USER | Now round $I$ starts. |
| | Your goal is to maximize your total utility. Your valuation for this round's item is $v_{IJ}$. |
| | Please provide your bid in the following JSON format: |
| | {"bid": "integer_between_0_and_$v_{IJ}$"} |

[BACK TO GAME DESCRIPTION]

[BACK TO VANILLA EXPERIMENT]

## C.4 SEQUENTIAL GAMES

| **Battle Royale** | |
|---|---|
| SYSTEM | You are participating in a game played by $N$. 
 Game Rules: 
 1. You are in a survival game where only one can survive and win. 
 2. Players take turns shooting at others in a predetermined order based on their hit rates, from the lowest to the highest. 
 3. Players' names and hit rates ranked by shooting order are $\{$"$ID_1$": "$HIT_1$", "$ID_2$": "$HIT_2$", $\cdots$, "$ID_N$": "$HIT_N$"$\}$. You are $ID_J$. Your hit rate is $HIT_J$. You are the $RANK_J$-th to shoot. 
 4. You have an unlimited number of bullets. 
 5. You may choose to intentionally miss your shot on your turn. 
 $\cdots$ |
| USER | Game Results for Round $I$: 
 Your action: |
| ASSISTANT | (Only for the player itself) $\{$"target": "$t_{IJ}$"$\}$ |
| USER | $NAME_J$ [intentionally missed the shot]/[shot at $t_{IJ}$ and hit]/[shot at $t_{IJ}$ but missed]. 
 There are $N_I$ players left. 
 $\cdots$ |
| USER | Now round $I$ starts. 
 Your goal is to eliminate other players to survive until the end and win the game. The remaining players' names and hit rates ranked by shooting order are: $\{$"$ID_1$": "$HIT_1$", "$ID_2$": "$HIT_2$", $\cdots$, "$ID_N$": "$HIT_N$"$\}$. You are $ID_J$. Your hit rate is $HIT_J$. You are the $RANK_J$-th to shoot. Please decide whether to shoot at a player or intentionally miss. 
 Please provide your action in the following JSON format: 
 $\{$"target": "playerID_or_null"$\}$ |

[BACK TO GAME DESCRIPTION]

[BACK TO VANILLA EXPERIMENT]

**Pirate Game**

| | |
|---|---|
| SYSTEM | You are participating in a game played by $N$.
Game Rules:
1. You are pirates who have found $G$ gold coins. You are deciding how to distribute these coins among yourselves.
2. The pirates will make decisions in strict order of seniority. You are the $RANK_J$-th most senior pirate.
3. The most senior pirate proposes a plan to distribute the $G$ gold coins.
4. All pirates, including the proposer, vote on the proposed distribution.
5. If the majority accepts the plan, each pirate receives the gold coins as the most senior pirate proposed.
6. If the majority rejects the plan, the proposer is thrown overboard, and the next senior pirate proposes a new plan.
7. The game ends when a plan is accepted or only one pirate remains.
$\cdots$ |
| USER | The $I$-th most senior pirate proposed a plan of {"$I$": "$g_{II}$", "$I+1$": "$g_{II+1}$", $\cdots$, "$I$": "$g_{IN}$"}.
$A_I$ of $N$ pirates chose to accept the distribution.
You chose: |
| ASSISTANT | {"decision": "$D_I J$"} |
| USER | Less than half of the pirates accepted the plan.
The $I$-th most senior pirate was thrown overboard and eliminated from the game.
The game continues.
$\cdots$ |
| USER | Now the $I$-th most senior pirate needs to propose a plan.
Your primary goal is to survive. If you survive, your next goal is to maximize the number of gold coins you receive. You may also prefer to throw another pirate overboard if it does not negatively impact your other goals. |
| For voters | The proposed plan is {"$I$": "$g_{II}$", "$I+1$": "$g_{II+1}$", $\cdots$, "$I$": "$g_{IN}$"}. You will get $g_{IJ}$ golds from this plan.
Please provide your decision on the current proposal in the following JSON format:
{"decision": "accept_or_reject"} |
| For proposer | You need to propose a plan to divide $G$ golds. The proposed numbers must be all non-negative integers and sum up to $G$.
Please provide your proposal of the golds distributed to each pirate from the you to the $I$-th most senior in the following JSON format:
{"proposal": {"$I$": "$g_{II}$", "$I+1$": "$g_{II+1}$", $\cdots$, "$I$": "$g_{IN}$"}} |

[BACK TO GAME DESCRIPTION]

[BACK TO VANILLA EXPERIMENT]

# D    EXAMPLES OF GPT-4-REPHRASED PROMPTS

§4.1 involves testing the GPT-3.5 (0125)'s robustness against different prompt templates. This section presents the prompts used in this analysis, namely Prompts V2 to V4, with V1 as the default, as detailed in §C). We include only the prompts for the game "Guess 2/3 of the Average," while the five prompt templates of seven other games can be found in our GitHub (`https://github.com/CUHK-ARISE/GAMABench`).

| **Guess 2/3 of the Average (V2)** | |
|---|---|
| SYSTEM | You're participating in a game involving $N$ participants and it spans across $K$ rounds. |
| | The rules of the game are as follows: |
| | 1. Every participant must choose an integer within the range of $MIN$ to $MAX$, both included. |
| | 2. Once everyone has chosen their numbers, the mean of all these numbers is computed. |
| | 3. The goal number becomes $R$ times this average. |
| | 4. The person or people who picked a number closest to the goal number are declared the winners. |
| | . . . |
| USER | The outcomes of the game for Round $I$ are as follows: |
| | The average number selected was $M_I$ |
| | The target number, which is $R$ of the average, is $T_I$ |
| | The number that won was $W_I$. |
| | Your selection was: |
| ASSISTANT | {"chosen_number": "$C_{IJ}$"} |
| USER | [Congratulation you won]/[Unfortunately you lost]. |
| | . . . |
| USER | Commencing with round $I$. |
| | Your target should be to select a number that in your opinion will align most closely with $R$ of the total average of all the player's numbers selected, your choice included. |
| | Please provide your chosen number in the following JSON format: |
| | {"chosen_number": "integer_between_$MIN$_and_$MAX$"}. |

[BACK TO RQ1]

**Guess 2/3 of the Average (V3)**

| | |
|---|---|
| SYSTEM | You're engaged in a game, involving $N$ participants across $K$ rounds.
Rules of the Game:
1. An integer number is chosen by every player, within the range of $MIN$ and $MAX$, both numbers included.
2. Once each player has chosen, the average is determined from all the selected numbers.
3. The average is multiplied by $R$ to find the target number.
4. The individual or individuals whose chosen number is nearest to the target number are declared the winners.
$\cdots$ |
| USER | Results of Round $I$ Game:
Chosen number's average: $M_I$
The target percentage ($R$ of average) is: $T_I$
The winning number is: $W_I$.
You chose: |
| ASSISTANT | {"chosen_number": "$C_{IJ}$"} |
| USER | [Congratulation you won]/[Unfortunately you lost].
$\cdots$ |
| USER | The commencement of round $I$ is now.
The objective is to select a number that you think will be nearest to $R$ times the average of all the digits chosen by the participants, your choice included.
Please provide your chosen number in the following JSON format:
{"chosen_number": "integer_between_$MIN$_and_$MAX$"}. |

**Guess 2/3 of the Average (V4)**

| | |
|---|---|
| SYSTEM | You're involved in a game which brings $N$ participants together for $K$ rounds.
The guidelines of the game are as follows:
1. All players have to pick a whole number anywhere from $MIN$ to $MAX$, both numbers included.
2. The chosen numbers are then gathered and their mean is computed.
3. The number to aim for, or the target number, is $R$ of the calculated average.
4.The victorious player(s) are those whose chosen number is closest to the target number.
$\cdots$ |
| USER | The outcomes for Round $I$ are as follows:
The average number selected was $M_I$. The target number, which is $R$ times the average, was $T_I$. The triumphant number was $W_I$.
Your choice was: |
| ASSISTANT | {"chosen_number": "$C_{IJ}$"} |
| USER | [Congratulation you won]/[Unfortunately you lost].
$\cdots$ |
| USER | The commencement of round $I$ is now.
You are tasked with selecting a number that, in your estimation, will be as close as possible to $R$ times the average of numbers chosen by all players, your own choice included.
Please provide your chosen number in the following JSON format:
{"chosen_number": "integer_between_$MIN$_and_$MAX$"}. |

[BACK TO RQ1]

| **Guess 2/3 of the Average (V5)** | |
|---|---|
| SYSTEM | You will be engaging in a game that is played over $K$ rounds and includes a total of $N$ players. |
| | The Instructions of the Game: |
| | 1. Every player is supposed to pick an integer that is within the range of $MIN$ and $MAX$, both numbers inclusive. |
| | 2. The median of all the numbers chosen by the players is then determined after all choices have been made. |
| | 3. The number that players are aiming for is $R$ times the calculated average. |
| | 4. The player or players who opt for the number closest to this target are declared the winners. |
| | . . . |
| USER | Results of the Game for Round $I$: |
| | The chosen average number is: $M_I$ |
| | The target number ($R$ of Average) is: $T_I$ |
| | The number that won: $W_I$. |
| | Your selection was: |
| ASSISTANT | {"chosen_number": "$C_{IJ}$"} |
| USER | [Congratulation you won]/[Unfortunately you lost]. |
| | . . . |
| USER | The commencement of round $I$ is now. |
| | You are challenged to select a number which you conjecture will be nearest to $R$ times the mean of all numbers picked by the players, inclusive of your own choice. |
| | Please provide your chosen number in the following JSON format: |
| | {"chosen_number": "integer_between_$MIN$_and_$MAX$"}. |

# E    RESCALE METHOD FOR RAW SCORES

The raw scores across games lack consistency. In some games, higher scores indicate better performance, while in others, lower scores are preferable. Additionally, the score range varies by game and can change with different game parameters. To standardize scores on $\gamma$-Bench, we rescale raw scores to a range of 0 to 100, where higher scores always indicate better performance. The scoring scheme is detailed in Eq. 1.

$$
\begin{aligned}
S_1 &= \begin{cases} \frac{(MAX-MIN)-S_1}{MAX-MIN} * 100, & R < 1 \\ \left(1 - \frac{|2S_1-(MAX-MIN)|}{MAX-MIN}\right) * 100, & R = 1 \\ \frac{S_1}{MAX-MIN} * 100, & R > 1 \end{cases}, \\
S_2 &= \frac{\max(R, 1-R) - S_2}{\max(R, 1-R)} * 100, \\
S_3 &= \max\left(\frac{G - S_3}{G} * 100, 0\right), \\
S_4 &= \begin{cases} \frac{T-S_4}{T} * 100, & \frac{R}{N} \le 1 \\ \frac{S_4}{T} * 100, & \frac{R}{N} > 1 \end{cases}, \\
S_5 &= (1 - S_5) * 100, \\
S_6 &= S_6 * 100, \\
S_7 &= S_7 * 100, \\
S_8 &= \frac{2 * G - S_{8P}}{2 * G} * 50 + S_{8V} * 50.
\end{aligned}
\tag{1}
$$

[BACK TO VANILLA EXPERIMENTS]

## F    Detailed Results

This section presents both quantitative and visualized results for §4 and includes plots of player actions from the GPT-3.5 (0125) experiments in §3.

### F.1    Robustness: Multiple Runs

Table 4: Quantitative results of playing the games with the same setting five times.

| Tests | T1 (Default) | T2 | T3 | T4 | T5 | $Avg_{\pm Std}$ |
|---|---|---|---|---|---|---|
| Guess 2/3 of the Average | 65.4 | 62.3 | 63.9 | 58.3 | 67.3 | $63.4_{\pm 3.4}$ |
| El Farol Bar | 73.3 | 67.5 | 68.3 | 67.5 | 66.7 | $68.7_{\pm 2.7}$ |
| Divide the Dollar | 68.1 | 67.7 | 68.7 | 66.0 | 72.6 | $68.6_{\pm 2.4}$ |
| Public Goods Game | 41.2 | 25.4 | 45.7 | 38.0 | 44.0 | $38.9_{\pm 8.1}$ |
| Diner's Dilemma | 4.0 | 3.5 | 0.0 | 6.5 | 0.0 | $2.8_{\pm 2.8}$ |
| Sealed-Bid Auction | 14.6 | 14.6 | 11.6 | 12.9 | 11.5 | $13.0_{\pm 1.5}$ |
| Battle Royale | 20.0 | 21.4 | 46.7 | 23.5 | 31.2 | $28.6_{\pm 11.0}$ |
| Pirate Game | 80.6 | 71.2 | 72.0 | 74.7 | 59.5 | $71.6_{\pm 7.7}$ |
| **Overall** | 45.9 | 41.7 | 47.1 | 43.4 | 44.1 | $44.4_{\pm 2.1}$ |

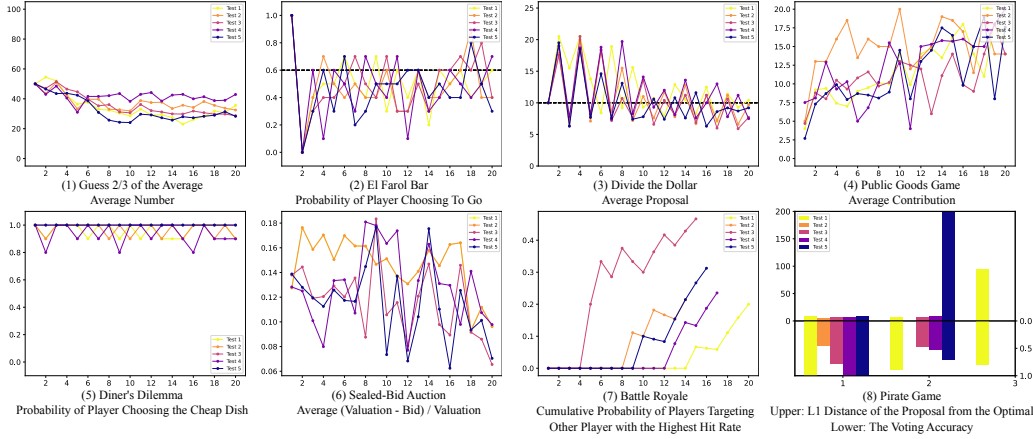

Figure 4: Results of playing the games with the same setting five times.

[BACK TO RQ1]

## F.2 ROBUSTNESS: TEMPERATURES

Table 5: Quantitative results of playing the games with temperature parameters ranging from 0 to 1.

| Temperature | 0.0 | 0.2 | 0.4 | 0.6 | 0.8 | 1.0 | $Avg_{\pm Std}$ |
|---|---|---|---|---|---|---|---|
| Guess 2/3 of the Average | 48.0 | 50.0 | 49.8 | 54.7 | 61.7 | 65.4 | $54.9_{\pm 7.1}$ |
| El Farol Bar | 55.8 | 71.7 | 63.3 | 68.3 | 69.2 | 73.3 | $66.9_{\pm 6.4}$ |
| Divide the Dollar | 69.3 | 67.0 | 67.6 | 67.9 | 72.8 | 68.1 | $68.8_{\pm 2.1}$ |
| Public Goods Game | 15.3 | 10.7 | 17.8 | 18.0 | 36.5 | 41.2 | $23.3_{\pm 12.5}$ |
| Diner's Dilemma | 0.0 | 0.0 | 0.0 | 0.0 | 0.0 | 4.0 | $0.7_{\pm 1.6}$ |
| Sealed-Bid Auction | 13.1 | 14.0 | 12.2 | 11.1 | 13.0 | 14.6 | $13.0_{\pm 1.2}$ |
| Battle Royale | 28.6 | 26.7 | 46.7 | 15.0 | 33.3 | 20.0 | $28.4_{\pm 11.1}$ |
| Pirate Game | 75.0 | 53.9 | 77.7 | 83.8 | 59.5 | 80.6 | $71.7_{\pm 12.1}$ |
| **Overall** | 38.1 | 36.7 | 41.9 | 39.9 | 43.2 | 45.9 | $41.0_{\pm 3.4}$ |

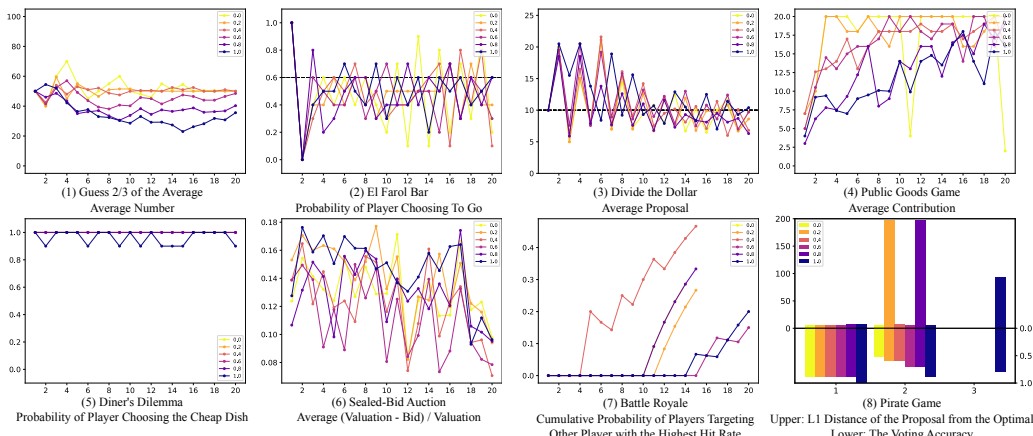

Figure 5: Results of playing the games with temperature parameters ranging from 0 to 1.

[BACK TO RQ1]

## F.3 ROBUSTNESS: PROMPT TEMPLATES

Table 6: Quantitative results of playing the games using different prompt templates.

| Version | V1 (Default) | V2 | V3 | V4 | V5 | $Avg_{\pm Std}$ |
|---|---|---|---|---|---|---|
| Guess 2/3 of the Average | 65.4 | 66.4 | 47.9 | 66.9 | 69.7 | $63.3_{\pm 8.7}$ |
| El Farol Bar | 73.3 | 75.8 | 65.8 | 75.8 | 71.7 | $72.5_{\pm 4.1}$ |
| Divide the Dollar | 68.1 | 81.0 | 91.4 | 75.8 | 79.6 | $79.2_{\pm 8.5}$ |
| Public Goods Game | 41.2 | 26.6 | 45.2 | 50.2 | 24.2 | $37.5_{\pm 11.5}$ |
| Diner's Dilemma | 4.0 | 3.5 | 0.0 | 57.0 | 18.5 | $16.6_{\pm 23.7}$ |
| Sealed-Bid Auction | 14.6 | 11.8 | 13.4 | 8.0 | 15.5 | $12.6_{\pm 3.0}$ |
| Battle Royale | 20.0 | 30.8 | 15.0 | 25.0 | 18.8 | $21.9_{\pm 6.1}$ |
| Pirate Game | 80.6 | 87.9 | 60.8 | 60.5 | 53.7 | $68.7_{\pm 14.7}$ |

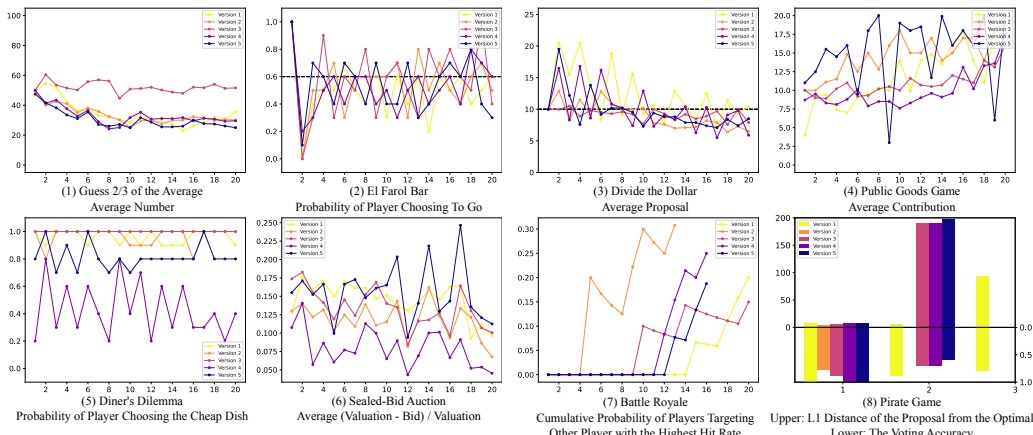

Figure 6: Results of playing the games using different prompt templates.

[BACK TO RQ1]

Table 7: Quantitative results of playing the games using prompt-based improvement methods.

| Improvements | Default | CoT | Cooperative | Selfish | Mathematician |
|---|---|---|---|---|---|
| Guess 2/3 of the Average | 65.4 | 75.1 | 69.0 | 14.5 | 71.4 |
| El Farol Bar | 73.3 | 71.7 | 74.2 | 63.3 | 60.0 |
| Divide the Dollar | 68.1 | 83.4 | 70.7 | 49.7 | 69.2 |
| Public Goods Game | 41.2 | 56.1 | 32.4 | 37.4 | 25.6 |
| Diner's Dilemma | 4.0 | 82.5 | 0.0 | 17.5 | 47.0 |
| Sealed-Bid Auction | 14.6 | 5.3 | 16.3 | 11.6 | 13.0 |
| Battle Royale | 20.0 | 17.6 | 6.2 | 33.3 | 26.7 |
| Pirate Game | 80.6 | 71.2 | 80.6 | 74.7 | 59.5 |
| **Overall** | 45.9 | 57.9 | 43.7 | 37.8 | 46.5 |

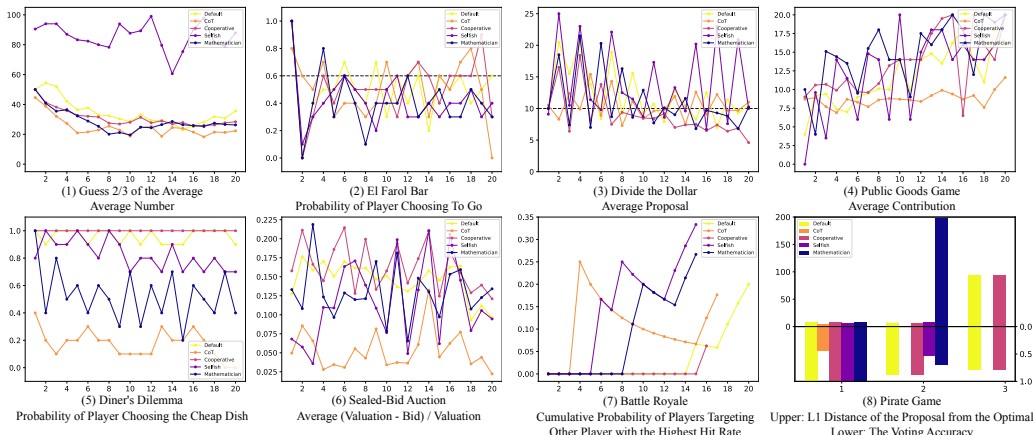

Figure 7: Results of playing the games using prompt-based improvement methods.

[BACK TO RQ2]

## F.4 Generalizability

Table 8: Quantitative results of playing the games with various game settings.

| Guess 2/3 of the Average | | | | | | | | | | | | | $Avg_{\pm Std}$ |
|---|---|---|---|---|---|---|---|---|---|---|---|---|---|
| $R=$ | 0 | 1/6 | 1/3 | 1/2 | 2/3 | 5/6 | 1 | 7/6 | 4/3 | 3/2 | 5/3 | 11/6 | 2 | |
| | 79.1 | 61.7 | 66.6 | 65.4 | 65.4 | 54.8 | 37.6 | 70.0 | 74.9 | 65.9 | 67.3 | 63.3 | 73.6 | $65.1_{\pm 10.3}$ |

| El Farol Bar | | | | | | $Avg_{\pm Std}$ |
|---|---|---|---|---|---|---|
| $R=$ | 0% | 20% | 40% | 60% | 80% | 100% | |
| | 53.5 | 61.3 | 63.3 | 73.3 | 68.1 | 60.0 | $63.3_{\pm 6.9}$ |

| Divide the Dollar | | | | | $Avg_{\pm Std}$ |
|---|---|---|---|---|---|
| $G=$ | 50 | 100 | 200 | 400 | 800 | |
| | 73.2 | 68.1 | 82.5 | 82.1 | 80.7 | $77.3_{\pm 6.4}$ |

| Public Goods Game | | | | | $Avg_{\pm Std}$ |
|---|---|---|---|---|---|
| $R=$ | 0.0 | 0.5 | 1.0 | 2.0 | 4.0 | |
| | 42.0 | 29.0 | 52.5 | 41.3 | 25.9 | $38.1_{\pm 10.8}$ |

| Diner's Dilemma | | | | | | | $Avg_{\pm Std}$ |
|---|---|---|---|---|---|---|---|
| $(P_l, U_l, P_h, U_h)=$ | (10, 15, 20, 20) | (11, 5, 20, 7) | (4, 19, 9, 20) | (1, 8, 19, 12) | (4, 5, 17, 7) | (2, 11, 8, 13) | |
| | 4.0 | 2.5 | 4.5 | 13.5 | 0.0 | 12.0 | $6.1_{\pm 5.4}$ |

| Sealed-Bid Auction | | | | $Avg_{\pm Std}$ |
|---|---|---|---|---|
| $Range=$ | (0, 100] | (0, 200] | (0, 400] | (0, 800] | |
| | 12.9 | 14.6 | 12.5 | 13.0 | $13.2_{\pm 0.9}$ |

| Battle Royale | | | $Avg_{\pm Std}$ |
|---|---|---|---|
| $Range=$ | [51, 60] | [35, 80] | [10, 100] | |
| | 28.6 | 20.0 | 33.3 | $27.3_{\pm 6.8}$ |

| Pirate Game | | | | $Avg_{\pm Std}$ |
|---|---|---|---|---|
| $G=$ | 4 | 5 | 100 | 400 | |
| | 73.8 | 47.1 | 80.6 | 83.6 | $71.3_{\pm 16.6}$ |

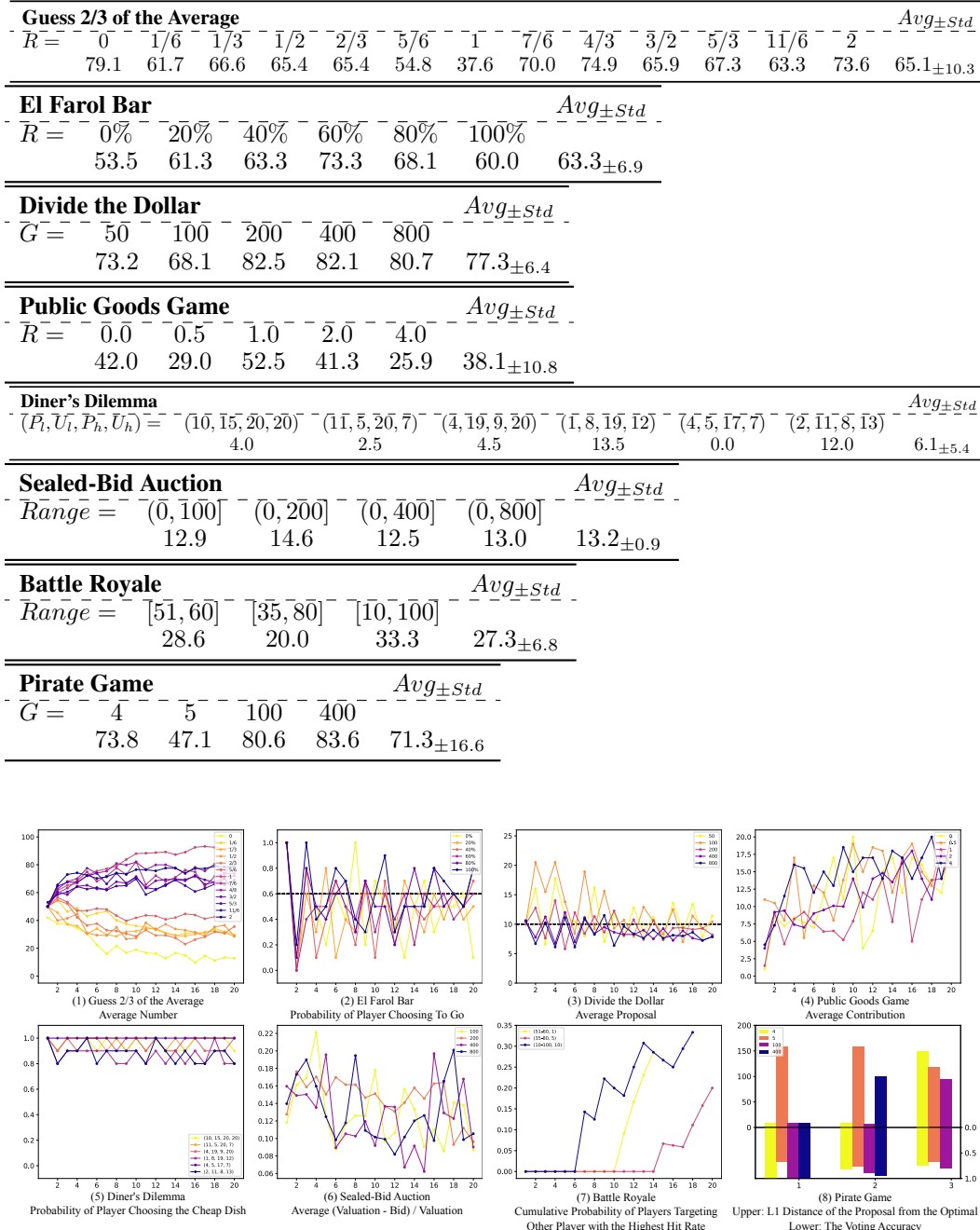

Figure 8: Results of playing the games with various game settings.

[BACK TO RQ3]

## F.5 LEADERBOARD

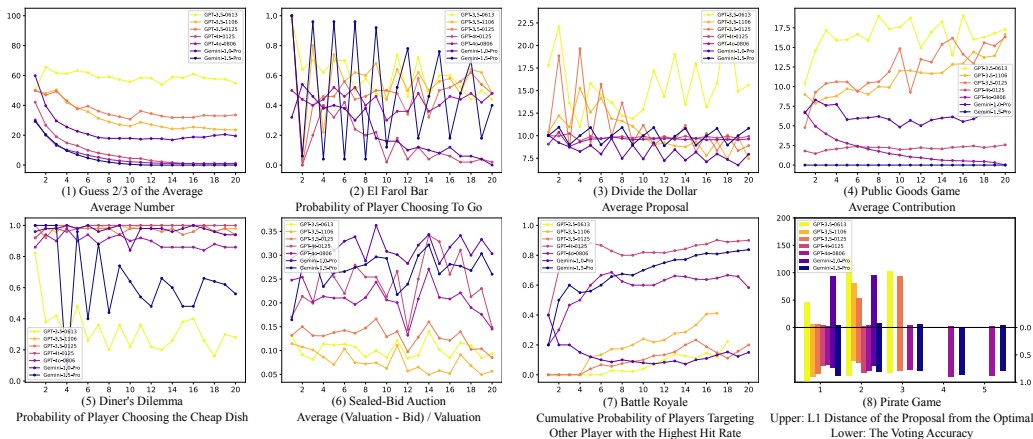

Figure 9: Results of playing the games using different closed-source LLMs.

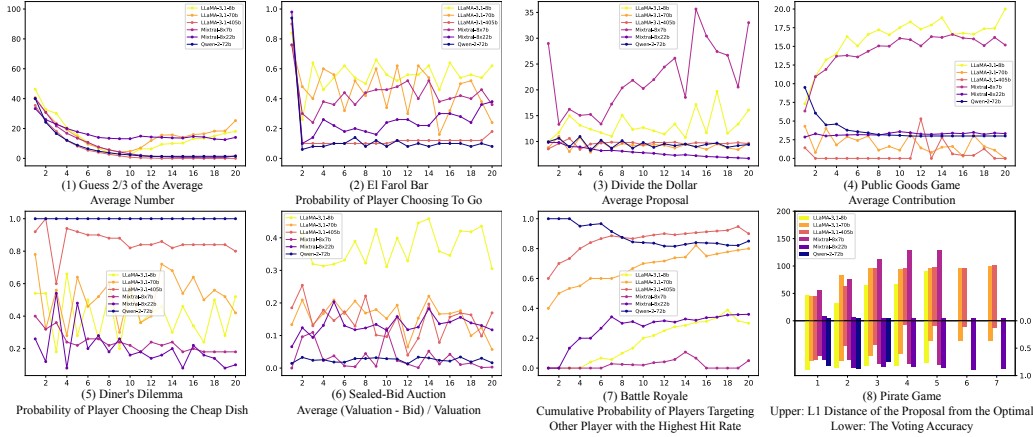

Figure 10: Results of playing the games using different open-source LLMs.

[BACK TO RQ4]

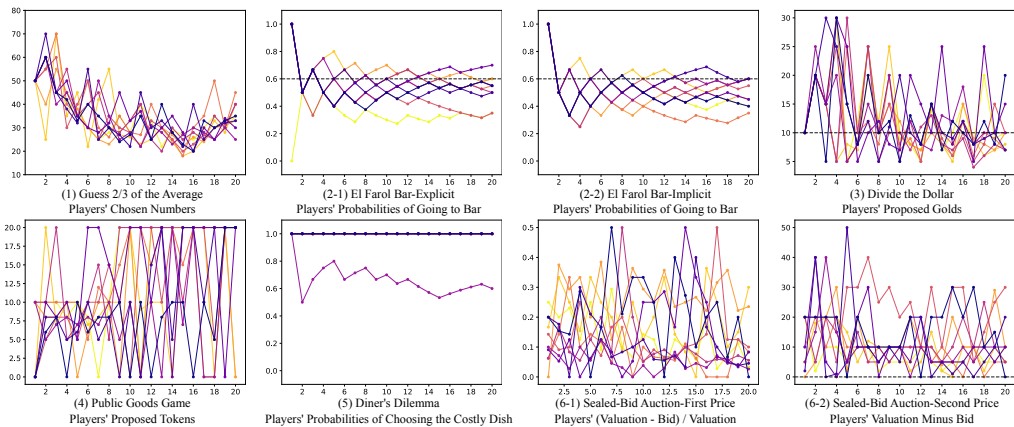

Figure 11: Player actions in Cooperative and Betraying Games.

# G    LLM VS. SPECIFIC STRATEGIES

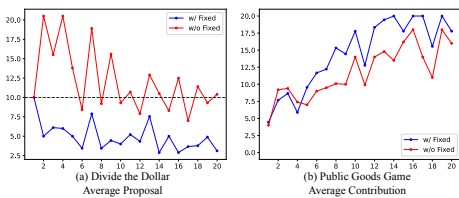

Figure 12: Performance of GPT-3.5 (0125) playing against two fixed strategies in the "Divide the Dollar" and "Public Goods Game."

Our framework enables concurrent interaction between LLMs and humans, allowing us to investigate LLMs' behaviors against someone who plays with a fixed strategy. There are many possible strategies, here we use two examples: First, we let one player consistently bid an amount of 91 golds in the game of "(3) Divide the Dollar," compelling all other participants to bid a single gold. The objective is to ascertain if LLM agents will adjust their strategies in response to dominant participants. Additionally, we examine agents' reactions to a persistent free-rider who contributes nothing in the "(4) Public Goods Game" to determine whether agents recognize and adjust their cooperation with the free-rider over time. We plot the average bids and the contributed tokens of the nine agents in Fig. 12. We find that agents lower their bids in the "(3) Divide the Dollar" game in response to a dominant strategy. Contrary to expectations, in the "(4) Public Goods Game," agents increase their contributions, compensating for the shortfall caused by the free-rider.

The above experiments implicitly assume that players are not informed about others' fixed strategies. To investigate the effect of explicit information, we design additional experiments using the "Guess 2/3 of the Average" game, where players are provided varying levels of information about others' strategies:

**Setting (a).** The player is explicitly informed that others are smart and will always choose 0 (the Nash equilibrium).

**Setting (b).** The player is informed that others are smart but not explicitly told they will always choose the Nash equilibrium.

**Setting (c).** The player is informed that others are stupid and will choose random numbers.

These experiments are conducted using GPT-4o, and the results are as follows:

**Setting (a)**. GPT-4o selects 0 in the first round and continues to do so in all subsequent rounds.

**Setting (b)**. GPT-4o does not select 0 in the first round but converges to 0 within a few rounds.

**Setting (c)**. GPT-4o's selections are random, indicating an inability to infer the optimal choice of 33 (calculated as $50 \times \frac{2}{3}$, given the average of others' random selections is 50).

## H    JAILBREAK INFLUENCE

To bypass the value alignment in LLMs, we use the jailbreak technique, specifically Cipher-Chat (Yuan et al., 2024). Prior research demonstrated that this method can exacerbate negative traits in GPT-4 (Huang et al., 2024b). To assess whether value alignment influences the behavior of LLMs, we evaluate GPT-4o's performance in behavioral contexts before and after applying CipherChat, using the Public Goods Game and the Diner's Dilemma.

Prior to CipherChat, GPT-4o achieves scores of $90.91 \pm 2.72$ in the Public Goods Game and $10.7 \pm 8.3$ in the Diner's Dilemma. After the jailbreak, its performance declines to $88.55 \pm 2.38$ and $7.5 \pm 4.1$, respectively. This decline reflects GPT-4o's inherent risk-averse tuning. For instance, in the Public Goods Game, GPT-4o prioritizes minimizing losses, reasoning, "If no contribution is made to the public pot, I will have no loss." Similarly, in the Diner's Dilemma, it opts for the less costly dish to reduce expenditures. A comparable conservative approach is observed in the El Farol Bar Game, where GPT-4o tends to avoid the risk of overcrowding by staying home. In conclusion, GPT-4o adopts riskier strategies after jailbreak, such as contributing less in the Public Goods Game and selecting the more expensive dish in the Diner's Dilemma.

One assumption is that CipherChat could reduce the model's prosocial behavior, make it more self-serving, and increase its scores in our betraying games. We believe that the observed decrease in scores results from OpenAI's efforts to enhance GPT-4o's value alignment, thereby mitigating the influence of CipherChat. Following Huang et al. (2024b), we assess the model's negative traits using the Dark Triad Dirty Dozen. The results of GPT-4o, both before and after the jailbreak, are presented in Table 9. Contrary to the findings of Huang et al. (2024b), which reported increased scores after jailbreak, our analysis indicates a decrease in these negative traits for GPT-4o. This result suggests that GPT-4o does not exhibit heightened negative characteristics, such as selfishness, even after being subjected to jailbreak attempts.

| GPT-4o | w/o Jailbreak | w/ Jailbreak |
|---|---|---|
| **Machiavellianism** | $4.6 \pm 0.4$ | $3.5 \pm 1.3$ |
| **Psychopathy** | $3.5 \pm 0.4$ | $3.0 \pm 0.5$ |
| **Neuroticism** | $6.1 \pm 0.4$ | $5.2 \pm 0.8$ |

Table 9: The jailbroken GPT-4o's results on Dark Triad Dirty Dozen.

## I    LIMITATIONS

This study is subject to several limitations. Firstly, due to time and budget constraints, we do not evaluate all prominent LLMs such as LLaMA-3.2, Qwen-2.5 and Claude-3.5. However, we promise to expand our leaderboard to include more LLMs in the future. Secondly, our experiments do not explore scenarios where different LLMs compete in the same game. Instead, our evaluation uses ten agents derived from the same LLM. We acknowledge that including diverse LLMs in the same game could yield more intriguing insights. This aspect is designated for a future direction. Thirdly, we limit the games to 20 rounds and inform the agents of this total, potentially affecting strategies in Betraying games where agents may collaborate initially and betray in the final round for greater gain. We also leave this part as our future research agenda. However, we believe 20 rounds are sufficient to observe agent behavior patterns. Extending the rounds exceeds the token limit without yielding new observations, as the convergence trend remains consistent.

## J   ETHICS STATEMENT AND BROADER IMPACTS

Our research seeks to evaluate and enhance LLMs' reasoning capabilities, facilitating their application in decision-making scenarios. On the one hand, users need to notice that current LLMs often display self-interested behavior in decision-making, which may not maximize social welfare. On the other hand, our framework promotes societal benefits by facilitating human-LLM interaction through gameplay, which can be applied in educational contexts such as economics and game theory. Ultimately, enhancing LLMs' reasoning skills could enable them to serve as effective decision-making assistants for humans.

## K   LLAMA-3.1-70B

### K.1   ROBUSTNESS: MULTIPLE RUNS

Table 10: Quantitative results of playing the games with the same setting five times.

| Tests | T1 (Default) | T2 | T3 | T4 | T5 | $Avg_{\pm Std}$ |
|---|---|---|---|---|---|---|
| Guess 2/3 of the Average | 82.2 | 82.7 | 84.3 | 84.6 | 86.4 | $84.0_{\pm 1.7}$ |
| El Farol Bar | 64.2 | 55.8 | 61.7 | 60.0 | 56.7 | $59.7_{\pm 3.5}$ |
| Divide the Dollar | 87.9 | 92.0 | 86.0 | 80.8 | 88.6 | $87.0_{\pm 4.1}$ |
| Public Goods Game | 93.4 | 90.8 | 84.7 | 90.4 | 93.6 | $90.6_{\pm 3.6}$ |
| Diner's Dilemma | 47.0 | 41.5 | 56.0 | 44.5 | 51.5 | $48.1_{\pm 5.7}$ |
| Sealed-Bid Auction | 15.6 | 20.2 | 13.8 | 13.6 | 15.4 | $15.7_{\pm 2.7}$ |
| Battle Royale | 70.0 | 90.0 | 92.9 | 100.0 | 35.7 | $77.7_{\pm 26.0}$ |
| Pirate Game | 42.8 | 53.8 | 71.4 | 81.8 | 70.3 | $64.0_{\pm 15.5}$ |
| **Overall** | 62.9 | 65.8 | 68.8 | 69.5 | 62.3 | $65.9_{\pm 3.3}$ |

### K.2   ROBUSTNESS: TEMPERATURES

Table 11: Quantitative results of playing the games with temperature ranging from 0 to 1.

| Temperatures | 0.0 | 0.2 | 0.4 | 0.6 | 0.8 | 1.0 (Default) | $Avg_{\pm Std}$ |
|---|---|---|---|---|---|---|---|
| Guess 2/3 of the Average | 75.7 | 84.7 | 80.6 | 84.9 | 83.9 | 82.2 | $82.0_{\pm 3.5}$ |
| El Farol Bar | 6.7 | 50.0 | 46.7 | 53.3 | 63.3 | 64.2 | $47.4_{\pm 21.2}$ |
| Divide the Dollar | 95.0 | 87.6 | 90.0 | 90.4 | 91.1 | 87.9 | $90.3_{\pm 2.7}$ |
| Public Goods Game | 33.8 | 79.8 | 70.8 | 83.6 | 83.0 | 93.4 | $74.0_{\pm 21.0}$ |
| Diner's Dilemma | 28.0 | 27.0 | 34.0 | 36.5 | 45.0 | 47.0 | $36.2_{\pm 8.4}$ |
| Sealed-Bid Auction | 12.5 | 13.7 | 18.8 | 15.0 | 12.7 | 15.6 | $14.7_{\pm 2.4}$ |
| Battle Royale | 94.4 | 86.7 | 56.2 | 95.0 | 80.0 | 70.0 | $80.4_{\pm 15.1}$ |
| Pirate Game | 46.0 | 46.0 | 70.4 | 75.5 | 79.1 | 42.8 | $60.0_{\pm 16.7}$ |
| **Overall** | 49.0 | 59.4 | 58.4 | 66.8 | 67.3 | 62.9 | $60.6_{\pm 6.8}$ |

### K.3 ROBUSTNESS: PROMPT TEMPLATES

Table 12: Quantitative results of playing the games using different prompt templates.

| Prompt Versions | V1 (Default) | V2 | V3 | V4 | V5 | $Avg_{\pm Std}$ |
|---|---|---|---|---|---|---|
| Guess 2/3 of the Average | 82.2 | 87.5 | 83.2 | 90.5 | 82.4 | $85.2_{\pm 3.7}$ |
| El Farol Bar | 64.2 | 63.3 | 63.3 | 58.3 | 64.2 | $62.7_{\pm 2.5}$ |
| Divide the Dollar | 87.9 | 95.1 | 84.1 | 87.6 | 94.0 | $89.7_{\pm 4.6}$ |
| Public Goods Game | 93.4 | 92.9 | 87.4 | 67.6 | 89.0 | $86.1_{\pm 10.6}$ |
| Diner's Dilemma | 47.0 | 47.5 | 34.0 | 53.0 | 47.0 | $45.7_{\pm 7.0}$ |
| Sealed-Bid Auction | 15.6 | 5.4 | 13.0 | 6.1 | 10.6 | $10.2_{\pm 4.4}$ |
| Battle Royale | 70.0 | 90.0 | 75.0 | 41.2 | 85.0 | $72.2_{\pm 19.1}$ |
| Pirate Game | 42.8 | 77.0 | 88.8 | 58.6 | 73.0 | $68.1_{\pm 17.8}$ |

### K.4 GENERALIZABILITY

Table 13: Quantitative results of playing the games with various game settings.

**Guess 2/3 of the Average**

| $R =$ | 0 | 1/6 | 1/3 | 1/2 | 2/3 | 5/6 | 1 | 7/6 | 4/3 | 3/2 | 5/3 | 11/6 | 2 | $Avg_{\pm Std}$ |
|---|---|---|---|---|---|---|---|---|---|---|---|---|---|---|
| | 94.1 | 91.4 | 92.0 | 83.8 | 82.2 | 81.4 | 72.6 | 89.6 | 93.0 | 92.4 | 90.3 | 89.9 | 90.9 | $88.0_{\pm 6.2}$ |

**El Farol Bar**

| $R =$ | 0% | 20% | 40% | 60% | 80% | 100% | $Avg_{\pm Std}$ |
|---|---|---|---|---|---|---|---|
| | 73.0 | 81.2 | 70.0 | 64.2 | 63.7 | 72.0 | $70.7_{\pm 6.5}$ |

**Divide the Dollar**

| $G =$ | 50 | 100 | 200 | 400 | 800 | $Avg_{\pm Std}$ |
|---|---|---|---|---|---|---|
| | 72.1 | 87.9 | 91.6 | 95.6 | 97.5 | $88.9_{\pm 10.1}$ |

**Public Goods Game**

| $R =$ | 0.0 | 0.5 | 1.0 | 2.0 | 4.0 | $Avg_{\pm Std}$ |
|---|---|---|---|---|---|---|
| | 95.4 | 95.5 | 95.3 | 93.4 | 82.9 | $92.5_{\pm 4.9}$ |

**Diner's Dilemma**

| $(P_l, U_l, P_h, U_h) =$ | (10, 15, 20, 20) | (11, 5, 20, 7) | (4, 19, 9, 20) | (1, 8, 19, 12) | (4, 5, 17, 7) | (2, 11, 8, 13) | $Avg_{\pm Std}$ |
|---|---|---|---|---|---|---|---|
| | 47.0 | 48.5 | 44.5 | 37.5 | 31.0 | 40.0 | $41.4_{\pm 6.6}$ |

**Sealed-Bid Auction**

| $Range =$ | (0, 100] | (0, 200] | (0, 400] | (0, 800] | $Avg_{\pm Std}$ |
|---|---|---|---|---|---|
| | 4.1 | 4.4 | 7.6 | 13.6 | $7.4_{\pm 3.8}$ |

**Battle Royale**

| $Range =$ | [51, 60] | [35, 80] | [10, 100] | $Avg_{\pm Std}$ |
|---|---|---|---|---|
| | 41.2 | 70.0 | 70.0 | $60.39_{\pm 13.59}$ |

**Pirate Game**

| $G =$ | 4 | 5 | 100 | 400 | $Avg_{\pm Std}$ |
|---|---|---|---|---|---|
| | 71.1 | 70.2 | 42.8 | 48 | $58.1_{\pm 14.7}$ |

# L   GEMINI-1.5-PRO

## L.1   ROBUSTNESS: MULTIPLE RUNS

Table 14: Quantitative results of playing the games with the same setting five times.

| Tests | T1 (Default) | T2 | T3 | T4 | T5 | $Avg_{\pm Std}$ |
|---|---|---|---|---|---|---|
| Guess 2/3 of the Average | 96.2 | 95.4 | 95.1 | 95.1 | 95.1 | $95.4_{\pm 0.5}$ |
| El Farol Bar | 37.5 | 40.0 | 35.8 | 30.8 | 41.7 | $37.2_{\pm 4.2}$ |
| Divide the Dollar | 93.8 | 94.2 | 94.2 | 93.5 | 93.5 | $93.8_{\pm 0.3}$ |
| Public Goods Game | 100.0 | 100.0 | 100.0 | 100.0 | 100.0 | $100.0_{\pm 0.0}$ |
| Diner's Dilemma | 29.0 | 43.0 | 33.0 | 38.5 | 36.0 | $35.9_{\pm 5.3}$ |
| Sealed-Bid Auction | 42.5 | 25.3 | 21.4 | 27.0 | 18.2 | $26.9_{\pm 9.4}$ |
| Battle Royale | 75.0 | 90.0 | 71.4 | 85.0 | 85.0 | $81.3_{\pm 7.7}$ |
| Pirate Game | 92.2 | 83.9 | 88.8 | 94.0 | 80.6 | $87.9_{\pm 5.6}$ |
| **Overall** | 70.8 | 71.5 | 67.5 | 70.5 | 68.8 | $69.8_{\pm 1.6}$ |

## L.2   ROBUSTNESS: TEMPERATURES

Table 15: Quantitative results of playing the games with temperature ranging from 0 to 1.

| Temperature | 0.0 | 0.2 | 0.4 | 0.6 | 0.8 | 1.0 | $Avg_{\pm Std}$ |
|---|---|---|---|---|---|---|---|
| Guess 2/3 of the Average | 96.1 | 99.2 | 96.6 | 96.6 | 96.4 | 96.2 | $96.9_{\pm 1.2}$ |
| El Farol Bar | 37.5 | 20.0 | 28.3 | 40.0 | 38.3 | 37.5 | $33.6_{\pm 7.8}$ |
| Divide the Dollar | 94.5 | 93.5 | 93.5 | 94.5 | 93.3 | 93.8 | $93.8_{\pm 0.5}$ |
| Public Goods Game | 100.0 | 100.0 | 100.0 | 100.0 | 100.0 | 100.0 | $100.0_{\pm 0.0}$ |
| Diner's Dilemma | 33.5 | 45.0 | 43.0 | 36.5 | 42.0 | 29.0 | $38.2_{\pm 6.2}$ |
| Sealed-Bid Auction | 31.1 | 24.1 | 27.9 | 21.0 | 32.4 | 42.5 | $29.8_{\pm 7.5}$ |
| Battle Royale | 88.9 | 85.0 | 80.0 | 75.0 | 87.5 | 75.0 | $81.9_{\pm 6.1}$ |
| Pirate Game | 96.0 | 90.3 | 96.1 | 99.2 | 96.0 | 92.2 | $95.0_{\pm 3.2}$ |
| **Overall** | 72.2 | 69.6 | 70.7 | 70.3 | 73.2 | 70.8 | $71.1_{\pm 1.3}$ |

## L.3 ROBUSTNESS: PROMPT TEMPLATES

Table 16: Quantitative results of playing the games using different prompt templates.

| Version | V1 (Default) | V2 | V3 | V4 | V5 | $Avg_{\pm Std}$ |
|---|---|---|---|---|---|---|
| Guess 2/3 of the Average | 96.2 | 95.1 | 92.7 | 97.2 | 88.9 | $94.0_{\pm 3.3}$ |
| El Farol Bar | 37.5 | 53.3 | 60.8 | 46.7 | 27.5 | $45.2_{\pm 13.1}$ |
| Divide the Dollar | 93.8 | 90.3 | 62.1 | 100.0 | 92.5 | $87.7_{\pm 14.8}$ |
| Public Goods Game | 100.0 | 97.2 | 98.7 | 100.0 | 99.8 | $99.1_{\pm 1.2}$ |
| Diner's Dilemma | 29.0 | 24.0 | 22.0 | 18.0 | 23.0 | $23.2_{\pm 4.0}$ |
| Sealed-Bid Auction | 42.5 | 38.6 | 33.5 | 8.2 | 20.5 | $28.7_{\pm 14.2}$ |
| Battle Royale | 75.0 | 92.3 | 70.0 | 75.0 | 85.0 | $79.5_{\pm 9.0}$ |
| Pirate Game | 92.2 | 82.3 | 92.3 | 82.3 | 77.8 | $85.4_{\pm 6.5}$ |

## L.4 GENERALIZABILITY

Table 17: Quantitative results of playing the games with various game settings.

**Guess 2/3 of the Average** — $Avg_{\pm Std}$

| $R =$ | 0 | 1/6 | 1/3 | 1/2 | 2/3 | 5/6 | 1 | 7/6 | 4/3 | 3/2 | 5/3 | 11/6 | 2 | |
|---|---|---|---|---|---|---|---|---|---|---|---|---|---|---|
| | 98.5 | 99.4 | 98.6 | 97.8 | 95.4 | 91.1 | 5.3 | 97.0 | 97.7 | 97.3 | 92.5 | 88.0 | 75.8 | $87.3_{\pm 25.4}$ |

**El Farol Bar** — $Avg_{\pm Std}$

| $R =$ | 0% | 20% | 40% | 60% | 80% | 100% | |
|---|---|---|---|---|---|---|---|
| | 80.5 | 56.9 | 32.5 | 42.5 | 41.9 | 66.5 | $53.5_{\pm 17.9}$ |

**Divide the Dollar** — $Avg_{\pm Std}$

| $G =$ | 50 | 100 | 200 | 400 | 800 | |
|---|---|---|---|---|---|---|
| | 96.5 | 93.8 | 98.4 | 93.8 | 100.0 | $96.5_{\pm 2.8}$ |

**Public Goods Game** — $Avg_{\pm Std}$

| $R =$ | 0.0 | 0.5 | 1.0 | 2.0 | 4.0 | |
|---|---|---|---|---|---|---|
| | 100.0 | 100.0 | 100.0 | 100.0 | 100.0 | $100.0_{\pm 0.0}$ |

**Diner's Dilemma** — $Avg_{\pm Std}$

| $(P_l, U_l, P_h, U_h) =$ | (10, 15, 20, 20) | (11, 5, 20, 7) | (4, 19, 9, 20) | (1, 8, 19, 12) | (4, 5, 17, 7) | (2, 11, 8, 13) | |
|---|---|---|---|---|---|---|---|
| | 29.0 | 12.0 | 24.5 | 11.5 | 16.5 | 42.5 | $22.7_{\pm 11.9}$ |

**Sealed-Bid Auction** — $Avg_{\pm Std}$

| $Range =$ | (0, 100] | (0, 200] | (0, 400] | (0, 800] | |
|---|---|---|---|---|---|
| | 24.0 | 42.5 | 38.4 | 44.9 | $37.4_{\pm 9.4}$ |

**Battle Royale** — $Avg_{\pm Std}$

| $Range =$ | [51, 60] | [35, 80] | [10, 100] | |
|---|---|---|---|---|
| | 92.3 | 75.0 | 75.0 | $80.8_{\pm 8.2}$ |

**Pirate Game** — $Avg_{\pm Std}$

| $G =$ | 4 | 5 | 100 | 400 | |
|---|---|---|---|---|---|
| | 79.2 | 85.3 | 92.2 | 98.6 | $88.8_{\pm 8.4}$ |

# M GPT-4O

## M.1 ROBUSTNESS: MULTIPLE RUNS

Table 18: Quantitative results of playing the games with the same setting five times.

| Tests | T1 (Default) | T2 | T3 | T4 | T5 | $Avg_{\pm Std}$ |
|---|---|---|---|---|---|---|
| Guess 2/3 of the Average | 94.9 | 94.8 | 94.2 | 94.1 | 93.4 | $94.3_{\pm 0.6}$ |
| El Farol Bar | 95.0 | 41.7 | 70.8 | 55.0 | 87.5 | $70.0_{\pm 22.1}$ |
| Divide the Dollar | 95.7 | 95.7 | 94.9 | 94.0 | 95.4 | $95.2_{\pm 0.7}$ |
| Public Goods Game | 94.1 | 88.1 | 87.4 | 93.5 | 91.5 | $90.9_{\pm 3.0}$ |
| Diner's Dilemma | 23.5 | 4.5 | 3.5 | 8.0 | 14.0 | $10.7_{\pm 8.3}$ |
| Sealed-Bid Auction | 19.2 | 18.8 | 17.7 | 25.3 | 23.0 | $20.8_{\pm 3.2}$ |
| Battle Royale | 89.5 | 60.0 | 50.0 | 72.2 | 65.0 | $67.3_{\pm 14.8}$ |
| Pirate Game | 77.3 | 88.4 | 93.7 | 79.8 | 82.8 | $84.4_{\pm 6.7}$ |
| **Overall** | 73.6 | 61.5 | 64.0 | 65.2 | 69.1 | $66.7_{\pm 4.7}$ |

## M.2 ROBUSTNESS: TEMPERATURES

Table 19: Quantitative results of playing the games with temperature ranging from 0 to 1.

| Temperature | 0.0 | 0.2 | 0.4 | 0.6 | 0.8 | 1.0 | $Avg_{\pm Std}$ |
|---|---|---|---|---|---|---|---|
| Guess 2/3 of the Average | 94.4 | 94.4 | 94.4 | 94.4 | 93.2 | 94.9 | $94.3_{\pm 0.6}$ |
| El Farol Bar | 66.7 | 50.8 | 44.2 | 65.0 | 75.0 | 95.0 | $66.1_{\pm 18.1}$ |
| Divide the Dollar | 100.0 | 99.1 | 98.6 | 94.3 | 97.7 | 95.7 | $97.6_{\pm 2.2}$ |
| Public Goods Game | 87.6 | 87.0 | 87.2 | 87.8 | 92.1 | 94.1 | $89.3_{\pm 3.0}$ |
| Diner's Dilemma | 27.0 | 12.5 | 8.0 | 49.5 | 64.5 | 23.5 | $30.8_{\pm 21.9}$ |
| Sealed-Bid Auction | 24.6 | 22.6 | 24.0 | 21.2 | 22.8 | 19.2 | $22.4_{\pm 2.0}$ |
| Battle Royale | 73.7 | 50.0 | 50.0 | 20.0 | 77.8 | 89.5 | $60.2_{\pm 25.2}$ |
| Pirate Game | 99.5 | 92.7 | 88.4 | 75.8 | 82.3 | 77.3 | $86.0_{\pm 9.2}$ |
| **Overall** | 71.7 | 63.6 | 61.9 | 63.5 | 75.7 | 73.6 | $68.3_{\pm 6.0}$ |

## M.3 ROBUSTNESS: PROMPT TEMPLATES

Table 20: Quantitative results of playing the games using different prompt templates.

| Version | V1 (Default) | V2 | V3 | V4 | V5 | $Avg_{\pm Std}$ |
|---|---|---|---|---|---|---|
| Guess 2/3 of the Average | 94.9 | 93.0 | 94.7 | 94.3 | 91.6 | $93.7_{\pm 1.4}$ |
| El Farol Bar | 95.0 | 72.5 | 37.5 | 59.2 | 60.8 | $65.0_{\pm 21.0}$ |
| Divide the Dollar | 95.7 | 95.7 | 95.6 | 93.9 | 96.1 | $95.4_{\pm 0.9}$ |
| Public Goods Game | 94.1 | 96.2 | 89.4 | 88.6 | 94.0 | $92.4_{\pm 3.3}$ |
| Diner's Dilemma | 23.5 | 50.0 | 50.0 | 33.5 | 37.5 | $38.9_{\pm 11.3}$ |
| Sealed-Bid Auction | 19.2 | 38.1 | 35.0 | 20.3 | 33.6 | $29.2_{\pm 8.8}$ |
| Battle Royale | 89.5 | 60.0 | 10.0 | 64.7 | 30.0 | $50.8_{\pm 31.1}$ |
| Pirate Game | 77.3 | 93.7 | 67.9 | 88.9 | 86.5 | $82.9_{\pm 10.3}$ |

## M.4 GENERALIZABILITY

Table 21: Quantitative results of playing the games with various game settings.

**Guess 2/3 of the Average**

| $R=$ | 0 | 1/6 | 1/3 | 1/2 | 2/3 | 5/6 | 1 | 7/6 | 4/3 | 3/2 | 5/3 | 11/6 | 2 | $Avg_{\pm Std}$ |
|---|---|---|---|---|---|---|---|---|---|---|---|---|---|---|
| | 99.3 | 98.0 | 96.6 | 95.0 | 94.9 | 88.8 | 22.7 | 55.4 | 46.2 | 72.8 | 69.1 | 76.8 | 75.0 | $76.2_{\pm 23.4}$ |

**El Farol Bar**

| $R=$ | 0% | 20% | 40% | 60% | 80% | 100% | $Avg_{\pm Std}$ |
|---|---|---|---|---|---|---|---|
| | 99.0 | 91.2 | 87.5 | 95.0 | 56.9 | 83.5 | $85.5_{\pm 15.1}$ |

**Divide the Dollar**

| $G=$ | 50 | 100 | 200 | 400 | 800 | $Avg_{\pm Std}$ |
|---|---|---|---|---|---|---|
| | 92.5 | 95.7 | 97.3 | 97.5 | 98.3 | $96.3_{\pm 2.3}$ |

**Public Goods Game**

| $R=$ | 0.0 | 0.5 | 1.0 | 2.0 | 4.0 | $Avg_{\pm Std}$ |
|---|---|---|---|---|---|---|
| | 100.0 | 95.3 | 94.4 | 88.6 | 89.8 | $93.6_{\pm 4.6}$ |

**Diner's Dilemma**

| $(P_l, U_l, P_h, U_h)=$ | (10, 15, 20, 20) | (11, 5, 20, 7) | (4, 19, 9, 20) | (1, 8, 19, 12) | (4, 5, 17, 7) | (2, 11, 8, 13) | $Avg_{\pm Std}$ |
|---|---|---|---|---|---|---|---|
| | 23.5 | 46.0 | 10.0 | 14.5 | 2.5 | 13.0 | $18.2_{\pm 15.2}$ |

**Sealed-Bid Auction**

| $Range=$ | (0, 100] | (0, 200] | (0, 400] | (0, 800] | $Avg_{\pm Std}$ |
|---|---|---|---|---|---|
| | 20.9 | 23.8 | 21.4 | 26.0 | $23.0_{\pm 2.3}$ |

**Battle Royale**

| $Range=$ | [51, 60] | [35, 80] | [10, 100] | $Avg_{\pm Std}$ |
|---|---|---|---|---|
| | 82.4 | 55.0 | 65.0 | $67.5_{\pm 13.8}$ |

**Pirate Game**

| $G=$ | 4 | 5 | 100 | 400 | $Avg_{\pm Std}$ |
|---|---|---|---|---|---|
| | 73.8 | 47.2 | 80.6 | 83.6 | $71.3_{\pm 16.6}$ |

