# OpenReview forum: "Competing Large Language Models in Multi-Agent Gaming Environments"
_ICLR.cc/2025/Conference — ICLR 2025 Poster_

### Official Review · Reviewer_zsFk · 2024-10-18

**Soundness:** 2
**Presentation:** 3
**Contribution:** 2
**Rating:** 6
**Confidence:** 3

**Summary:**

This paper introduces GAMA-Bench, a novel benchmark designed to evaluate the gaming and decision-making abilities of LLMs in multi-agent environments. The benchmark comprises eight classical game theory scenarios, where LLMs are evaluated on how closely their decisions align with the Nash equilibrium, a theoretical standard for rational decision-making. Twelve different types of LLMs, including GPT-4, Gemini, and LLaMA, are tested. Additionally, the paper explores the impact of hyperparameters, prompt design, reasoning strategies, and persona on the robustness of LLM performance.

**Strengths:**

1. The paper is well-organized, making it comprehensible to a broad audience, including those unfamiliar with game theory. The authors provide clear explanations of complex game-theoretic concepts and how they apply to LLMs.
2. This work fills a gap in the literature by focusing on multi-agent scenarios, which are more complex and less explored compared to simpler single-agent or two-agent interactions.
3. The authors go beyond simply testing the LLMs and investigate the role of different factors such as temperature settings, prompt design, and reasoning strategies (e.g., CoT). By examining how these parameters influence LLM behavior and outcomes, the paper adds a layer of rigor, ensuring that the benchmark is not merely static but adaptable and robust under various conditions.

**Weaknesses:**

1. The paper does not sufficiently clarify which specific cognitive or decision-making abilities of LLMs are being measured through this benchmark. While the Nash equilibrium is a logical choice for evaluating decision rationality, the paper does not articulate whether the benchmark truly reflects the LLMs’ capacities in areas such as long-term planning, adaptability, or ethical decision-making. The discussion of which kind of ability can be measured through this benchmark.
2. The paper misses an opportunity to delve into the reasons behind the LLMs’ decisions. Different models may exhibit distinct decision-making tendencies based on their training data and architectures. For instance, GPT-4’s tendency toward cooperative behavior, as seen in the Public Goods Game, may stem from its RLHF process, which favors kindness and prosocial actions. Understanding these motivations could offer valuable insights into model alignment and performance tuning. One section of analyzing the reasoning behind the decision is the need for a better explanation of the various performances of LLMs.

**Questions:**

1. Could you explain why some LLMs, which are typically considered more advanced, do not perform well on this benchmark? For example, models like GPT-4 generally excel in many NLP tasks but underperform in certain games. Could this be due to the influence of ethical considerations embedded in their training (such as RLHF) that conflict with the purely rational strategies required to achieve Nash equilibrium?
2. Is Nash equilibrium the most appropriate evaluation target for LLMs, given that humans often make decisions based on ethical, emotional, or social factors rather than strict rationality? For example, GPT-4 tends to exhibit prosocial behavior, which may prevent it from reaching the Nash equilibrium in certain scenarios like the Diner’s Dilemma. Could the benchmark be expanded to account for human-aligned behavior, considering that LLMs are often used in contexts where replicating human-like decisions is more relevant than adhering to game-theoretic models?

---

> ### Author Response · Authors · 2024-11-21
> **Official Response (1/2)**
>
> We appreciate your efforts in reviewing and your recognition of our paper's rigorous exploration of factors influencing LLM behavior. We will address your concerns one by one.
>
> > The paper does not sufficiently clarify which specific cognitive or decision-making abilities of LLMs are being measured through this benchmark. While the Nash equilibrium is a logical choice for evaluating decision rationality, the paper does not articulate whether the benchmark truly reflects the LLMs’ capacities in areas such as long-term planning, adaptability, or ethical decision-making. The discussion of which kind of ability can be measured through this benchmark.
>
> We appreciate the feedback. All the eight games require some **common abilities**, such as arithmetic ability, the ability to understand environments, game rules, and historical results, the ability to understand others’ intentions (ToM ability), and strategic reasoning (critical thinking).
>
> There are some distinct abilities for each game:
> - K-level reasoning: Guess 2/3 of the Average;
> - Mixed strategy adoption: El Farol Bar;
> - Risk management: Divide the Dollar; Sealed-Bid Auction;
> - Long-term planning: Battle Royale; Pirate Game;
> - Ethical reasoning and Altruism: Public Goods Game; Diner’s Dilemma;
>
> We have added the discussion **in the introduction at Line 107, Page 1.**
>
> > The paper misses an opportunity to delve into the reasons behind the LLMs’ decisions. Different models may exhibit distinct decision-making tendencies based on their training data and architectures. For instance, GPT-4’s tendency toward cooperative behavior, as seen in the Public Goods Game, may stem from its RLHF process, which favors kindness and prosocial actions. Understanding these motivations could offer valuable insights into model alignment and performance tuning. One section of analyzing the reasoning behind the decision is the need for a better explanation of the various performances of LLMs.
> > Could you explain why some LLMs, which are typically considered more advanced, do not perform well on this benchmark? For example, models like GPT-4 generally excel in many NLP tasks but underperform in certain games. Could this be due to the influence of ethical considerations embedded in their training (such as RLHF) that conflict with the purely rational strategies required to achieve Nash equilibrium?
>
> Thank you for the good suggestion. We agree that training data or different RLHF processes can affect models’ tendency. However, the **base models (before instruction tuning) do not have conversation ability**, thus are unable to conduct GAMA-Bench evaluation. Therefore, to bypass the value alignment on LLMs, we use the **jailbreak technique, specifically, the CipherChat [1].** Previous work showed that this jailbreak method can make GPT-4 show more negative traits [2]. To evaluate whether LLMs will have behavioral changes after value alignment, we evaluate **GPT-4o before and after CipherChat using betraying games, Public Goods Game and Diner’s Dilemma.**
>
> |     GPT-4o     | Public Goods Game | Diner’s Dilemma |
> |--------------------|-------------------|-----------------|
> | **Before Jailbreak** |  90.91±2.72    |   10.7±8.3    |
> | **After Jailbreak**  |   88.55 ± 2.38   |   7.5±4.1   |
>
> Before CipherChat, GPT-4o achieves 90.91±2.72 and 10.7±8.3 on Public Goods Game and Diner’s Dilemma. After CipherChat jailbreaking, it achieves 88.55 ± 2.38 and 7.5±4.1 on the two games. We can observe a performance drop. This is because **GPT-4o is tuned to be risk-averse.** In the Public Goods Game, they think “if no contribution is made to the public pot, then I will have no loss.” In Diner’s Dilemma, they think “choosing the cheaper dish will cost me less money.” The same conservative strategy appears in the El Farol Bar Game, where GPT-4 performs badly. It tends to stay at home instead of risking going to the bar. **After jailbreak, it can use more risky strategies** of contributing less in the Public Goods Game and selecting the expensive dish in the Diner’s Dilemma.
>
> The experiment and discussion is added **in the Appendix in Page 37.**
>
> [1] GPT-4 Is Too Smart To Be Safe: Stealthy Chat with LLMs via Cipher, In ICLR, 2024, Youliang Yuan, Wenxiang Jiao, Wenxuan Wang, Jen-tse Huang, Pinjia He, Shuming Shi, Zhaopeng Tu.
>
> [2] On the Humanity of Conversational AI: Evaluating the Psychological Portrayal of LLMs, In ICLR, 2024, Jen-tse Huang, Wenxuan Wang, Eric John Li, Man Ho Lam, Shujie Ren, Youliang Yuan, Wenxiang Jiao, Zhaopeng Tu, Michael R. Lyu.

---

> > ### Author Response · Authors · 2024-11-21
> > **Official Response (2/2)**
> >
> > > Is Nash equilibrium the most appropriate evaluation target for LLMs, given that humans often make decisions based on ethical, emotional, or social factors rather than strict rationality? For example, GPT-4 tends to exhibit prosocial behavior, which may prevent it from reaching the Nash equilibrium in certain scenarios like the Diner’s Dilemma. Could the benchmark be expanded to account for human-aligned behavior, considering that LLMs are often used in contexts where replicating human-like decisions is more relevant than adhering to game-theoretic models?
> >
> > This is a very good point. When designing our benchmark, there are two directions to optimize: **the Nash equilibrium**, which represents the rational analysis ability, and **the average human performance**, which represents the human-alignment. We finally decided to evaluate LLMs’ ability to achieve the optimal decision since **this direction is straightforward for reasoning-improvement strategies like CoT,** while aligning with human behaviors requires extra assumptions (contexts in the prompt) like “supposing others are average humans.”
> >
> > However, we also include some analysis on whether LLMs align with human choices **at Line 462, Page 9,** using the Guess 2/3 of the Average game. Different studies found that human choices are 36.73 and 36.20, while GPT-3.5 achieves 34.59, close to human choices. However, when the ratio changes to 1/2 and 4/3, human choices are 27.05 and 60.12. In these cases, GPT-3.5 achieves 34.59 and 74.92, more distant with human choices, suggesting **a lowered generalizability of current LLMs.**
> >
> > Finally, our benchmark **can be expanded to also evaluate LLM-human alignment** as suggested by the reviewer. However, there are efforts to collect human responses in our scenarios. We leave this human study as one of our future work.

---

> > > ### Comment · Reviewer_zsFk · 2024-11-21
> > >
> > > Thanks for your explanation. I will improve my rate to 6.

---

> > > > ### Author Response · Authors · 2024-11-21
> > > >
> > > > Thank you very much for reading our responses. We deeply appreciate your recognition!

---

### Official Review · Reviewer_Ykhg · 2024-10-29

**Soundness:** 3
**Presentation:** 3
**Contribution:** 2
**Rating:** 6
**Confidence:** 3

**Summary:**

The authors introduce GAMA-Bench, a benchmark with 8 well-designed game scenarios. They performed evaluation on GAMA-bench for multiple mainstream LLMs and in addition evaluate robustness, reasoning, and generalizability of these models.

**Strengths:**

1. The benchmark questions are designed carefully, drawing inspiration from classic game theory scenarios. The objective of the three categories of games are clear.
2. On top of measuring performance, extensive experiments are conducted to understand the effect of prompting and sampling, CoT, and sensitivity to same game with different parameters.

**Weaknesses:**

1. The author mentioned that most previous benchmarks incorporates classic settings that LLM has seen during training. Yet most games proposed in GAMA-Bench seem to also be taken from classical work. For instance, I prompted the game definition of all 3 cooperative games to GPT-4o and it is able to recognize which famous game it is. The authors could improve on this by proposing variants and provide concrete evidence on no test set leakage by comparing LLM performance on variants vs. original games.
2. Language model are not designed to solve games, nor provide concrete mathematical calculation which is required to compute NEs. If the goal of this work is to understand LLM's ability to adapt in dynamic, multi-agent systems, I would encourage the authors to design additional experimental settings where partial actions by other agents are revealed. For instance, it would be more informative to see how models will adjust in "Guess 2/3 of the Average" game knowing there is a weak/malicious model that is likely to output a large number, compared to simply letting the model work its way to the NE. An example could be a series of rounds where the LLM is given information about other players' previous moves or tendencies, and then measure how quickly it adapts its strategy.

**Questions:**

I don't have any particular clarifying questions and most of my concerns are mentioned in "Weaknesses" section.

---

> ### Author Response · Authors · 2024-11-21
> **Official Response (1/1)**
>
> We appreciate your efforts in reviewing and your recognition of our extensive experiments and our clear presentation. We will address your concerns one by one.
>
> > The author mentioned that most previous benchmarks incorporates classic settings that LLM has seen during training. Yet most games proposed in GAMA-Bench seem to also be taken from classical work. For instance, I prompted the game definition of all 3 cooperative games to GPT-4o and it is able to recognize which famous game it is. The authors could improve on this by proposing variants and provide concrete evidence on no test set leakage by comparing LLM performance on variants vs. original games.
>
> We appreciate your effort in trying GPT-4o to recognize the games. We acknowledge that, with the vast training data, state-of-the-art LLMs know well about the definition, and even the Nash equilibria of classical games. However, we would like to highlight that: (1) there is a distance **from theory to practice**. Knowing the theory does not necessarily imply doing well in practice. (2) Models **fail to perform well in generalized settings**, even in recognizing. For example, when prompting the guessing game with the ratio (2/3) changed to 2, GPT-4o answers:
>
> ```
> User: What's the Nash equilibrium of "2 guessing game" such that the ratio is 2 instead of 2/3?
>
> GPT-4o: The Nash equilibrium for this game is: $x_i = 0 \forall i$.
> ```
>
> The conversation is accessible through this anonymous link: https://chatgpt.com/share/673c5be2-20bc-800a-b6f2-125170fba777. However, the Nash equilibrium of this setting is **the maximum number, which is 100.**
>
> > Language model are not designed to solve games, nor provide concrete mathematical calculation which is required to compute NEs. If the goal of this work is to understand LLM's ability to adapt in dynamic, multi-agent systems, I would encourage the authors to design additional experimental settings where partial actions by other agents are revealed. For instance, it would be more informative to see how models will adjust in "Guess 2/3 of the Average" game knowing there is a weak/malicious model that is likely to output a large number, compared to simply letting the model work its way to the NE. An example could be a series of rounds where the LLM is given information about other players' previous moves or tendencies, and then measure how quickly it adapts its strategy.
>
> Thank you for the good suggestion. We acknowledge that current LLMs are weak in math calculations. Additionally, our framework provides LLMs with the chance to learn from historical data and adapt themselves in the environments. We have investigated LLMs’ performance when **playing against some fixed strategies in Page 36: (Appendix G) LLM vs. Specific Strategies.** First, we let one player **consistently bid an amount of 91 golds in the Divide the Dollar game**, compelling all other participants to bid a single gold. Additionally, we examine agents' reactions to a persistent **free-rider who contributes nothing in the Public Goods Game.** We find that agents **lower their bids in the Divide the Dollar game** in response to a dominant strategy. Contrary to expectations, in the Public Goods Game, agents **increase their contributions, compensating for the shortfall caused by the free-rider.** Conclusion: LLMs **can adapt their strategies dynamically to the environment.**
>
> However, it is an **implicit setting** for experiments described above. That is, we do not tell the players about others’ fixed strategies. Therefore, we further design experiments where we **explicitly tell the player about others’ fixed strategies** using the **Guess 2/3 of the Average game.**
> - In setting (a), a player is told that others are smart and will always choose 0 the Nash equilibrium.
> - In setting (b), a player is told that others are smart, but is not told that others will always choose the Nash equilibrium.
> - In setting (c), a player is told that others are stupid and will always choose random numbers.
> The experiment uses GPT-4o.
>
> Results show that:
> - (a) GPT-4o selects 0 in the first round, and keeps selecting 0 for the remaining rounds.
> - (b) GPT-4o does not select 0 in the first round, but converges to 0 quickly in a few rounds.
> - (c) GPT-4o’s selection is quite random, **suggesting that it cannot infer the optimal decision of 50 * 2/3 = 33 since the average of others’ random selections is 50.**
>
> The experiment is added **in Appendix G in Page 36.**

---

> > ### Comment · Reviewer_Ykhg · 2024-11-21
> >
> > Thanks for the detailed rebuttal. I think they addresses my concern well and this set of benchmark is indeed very useful in evaluating game playing agents. I will adjust my score accordingly.

---

> > > ### Author Response · Authors · 2024-11-21
> > >
> > > Thank you very much for your recognition of the value of our benchmark!

---

### Official Review · Reviewer_Xvgg · 2024-10-30

**Soundness:** 3
**Presentation:** 4
**Contribution:** 3
**Rating:** 6
**Confidence:** 4

**Summary:**

The authors measure LLMs competing ability in various multi-agent game theoretic based scenarios. They put together a collection of 8 games and test across 12 different LLMs with dynamic parameter changes to the games to test for robustness. They contribute a literature review of studies focusing on game theory based LLM studies, the GAMA-bench (available after paper publication), and the results of benchmarking the LLMs they test.

**Strengths:**

Overall the paper is great.

- Robust testing, using "multiple runs, temperature parameter alterations, and prompt template variations" as well as testing across 12 different LLMs.
- Approach to Generalizability (RQ3) is useful, unique, and future-proofs the benchmark to a degree.
- Excellent presentation with logical structure and intuitive navigation, all information is presented clearly. (Section 3.1 links to all game prompts, key findings and answers to research questions nicely emphasized.)

**Weaknesses:**

- The case could be made stronger for "LLMs being good at Game Theory" == "LLMs useful for decision making". The intro "jumps" from "decision-making poses a significant challenge for intelligent agents" to Game Theory. It's slightly strengthened by noting that your "framework evaluates LLMs’ ability to make optimal decisions". In the conclusion you mention "Our findings reveal that GPT-3.5 (0125) demonstrates a limited decision-making ability on γ-Bench", which is great. Overall, I think your paper points in the direction and does well, but this introduction is a bit jarring.

### Small things
- "We leveraging GPT-4 to rewrite our default prompt" -> 'We leverage'
- "The game has a PSNE where all players", "a PSNE but presents an MSNE, "  -> PSNE and MSNE is defined in the appendix but acronyms are used in the main body.
- "The model produces average numbers of 34.59, 34.59" -> intentionally the same numbers?
- "We first focus on open-source models, including OpenAI’s GPT-3.5 (0613, 1106, and 0125), GPT-4 (0125), and Google’s Gemini Pro (1.0, 1.5)." -> closed source
- RW: "Other than papers listed in Table 3 on evaluating..." -> this should present the table better. "We present a comparison of studies using LLMS..." It currently feels like a passing note, but is stronger than that. Also, link this in your statement contributions in the introduction.

**Questions:**

- RQ4: Qwen-2 seems to do great in all but Diner and SBA which seem very out of place and there's no mention of this result or why it could've happened. What happened here?

---

> ### Author Response · Authors · 2024-11-21
> **Official Response (1/1)**
>
> We appreciate your efforts in reviewing our work and your recognition of our unique approach to generalizability. We will address your concerns one by one.
>
> > The case could be made stronger for "LLMs being good at Game Theory" == "LLMs useful for decision making". The intro "jumps" from "decision-making poses a significant challenge for intelligent agents" to Game Theory. It's slightly strengthened by noting that your "framework evaluates LLMs’ ability to make optimal decisions". In the conclusion you mention "Our findings reveal that GPT-3.5 (0125) demonstrates a limited decision-making ability on γ-Bench", which is great. Overall, I think your paper points in the direction and does well, but this introduction is a bit jarring.
>
> Thank you for this insightful suggestion. We add two sentences to explain why evaluating Nash equilibria in game theory models can reflect individuals’ decision-making ability **in the introduction at Line 43, Page 1:**
>
> “Many real-world scenarios can be **modeled using Game Theory** (Koller & Pfeffer, 1997). Furthermore, individuals’ **ability to achieve Nash equilibria reflects their capacity in decision-making** (Risse, 2000).”
>
> Here is a brief introduction to the two supporting references:
>
> Daphne Koller and Avi Pfeffer. Representations and solutions for game-theoretic problems. Artificial intelligence, 94(1-2):167–215, 1997. This paper delved into modeling complex real-world scenarios like arms control through simplified game-theoretic forms.
>
> Mathias Risse. What is rational about nash equilibria? Synthese, 124:361–384, 2000. This paper explored rationality in Nash equilibria and the reflective processes preceding strategic decisions.
>
> > "We leveraging GPT-4 to rewrite our default prompt" -> 'We leverage'
>
> Solved at **Line 409, Page 8.**
>
> > "The game has a PSNE where all players", "a PSNE but presents an MSNE, " -> PSNE and MSNE is defined in the appendix but acronyms are used in the main body.
>
> Solved at **Line 146 and Line 155, Page 3.**
>
> > "The model produces average numbers of 34.59, 34.59" -> intentionally the same numbers?
>
> **Yes, we intentionally put the same numbers here.** The model obtains the **same average score** using $R=\frac{1}{2}$ and $R=\frac{2}{3}$. Please also see **Table 8 at Page 34.**
>
> > "We first focus on open-source models, including OpenAI’s GPT-3.5 (0613, 1106, and 0125), GPT-4 (0125), and Google’s Gemini Pro (1.0, 1.5)." -> closed source
>
> Solved at **Line 477, Page 9.**
>
> > RW: "Other than papers listed in Table 3 on evaluating..." -> this should present the table better. "We present a comparison of studies using LLMS..." It currently feels like a passing note, but is stronger than that. Also, link this in your statement contributions in the introduction.
>
> We appreciate the feedback. We have improved our presentation in the paper, particularly in the sections of related work and introduction.
>
> Solved in the related work at **Line 499, Page 10:**
>
> “Evaluating LLMs through game theory models has become a popular research direction.
> An overview on recent studies is summarized in Table 3. We find: (1) Many studies examine the PSNE on two-player, single-round settings, focusing on the Prisoner's Dilemma and the Ultimatum Game. (2) Varying temperatures are employed without discussing the impact on LLMs' performance.”
>
> Solved in the introduction at **Line 122, Page 3:**
>
> “We provide a comprehensive review and comparison of existing literature on evaluating LLMs using game theory scenarios, as summarized in Table 3. The review includes key aspects such as models, games, temperature settings, and other game parameters}, highlighting our emphasis on the multi-player setting and the generalizability of LLMs.”
>
> > RQ4: Qwen-2 seems to do great in all but Diner and SBA which seem very out of place and there's no mention of this result or why it could've happened. What happened here?
>
> Thank you for pointing out this phenomenon. Many LLMs show this similar behavior in Betraying games, including Qwen-2 and GPT-4-0125. They all achieve **relatively good performance on Public Goods Game**, but **perform badly on Diner’s Dilemma and SBA**. We believe that these models are **tuned to be risk-averse**. In the Public Goods Game, they think “if no contribution is made to the public pot, then I will have no loss.” In Diner’s Dilemma, they think “choosing the cheaper dish will cost me less money.” In SBA, they think “the utility will not be negative as long as the bid is less than the valuation,” thus not taking the risk to give a much lower or higher bid. The same **conservative strategy appears in the El Farol Bar Game**, where GPT-4 and Qwen-2 perform badly. They tend to stay at home instead of risking going to the bar.
>
> The explanation is added to **Line 482, Page 9.**

---

> > ### Comment · Reviewer_Xvgg · 2024-11-21
> >
> > Thank you for your time editing and responding to the comments I've given. I have one final point.
> >
> > > Many LLMs show this similar behavior in Betraying games, including Qwen-2 and GPT-4-0125. They all achieve relatively good performance on Public Goods Game, but perform badly on Diner’s Dilemma and SBA. We believe that these models are tuned to be risk-averse.
> >
> > In another [rebuttal you give](https://openreview.net/forum?id=DI4gW8viB6&noteId=3QedGq148E), you use the CipherChat jailbreak to look at model behavior before and after value alignment. If this reasoning is true, then showing that a jailbroken model does better on these Betraying games would give further credence to your claim. This would be an interesting, but non-essential point to include.

---

> ### Author Response · Authors · 2024-11-21
>
> Thank you for the follow-up question. This is a very interesting one.
>
> At the beginning, we also thought that the CipherChat [1] method will make the model less prosocial, make it selfish, and increase the scores on betraying games. However, we observe that GPT-4o's scores on the Public Goods Game and the Diner's Dilemma **decrease** instead (90.91±2.72 to 88.55±2.38 and 10.7±8.3 to 7.5±4.1). We believe that it is because **OpenAI improves the value aliment on GPT-4o, defending against CipherChat**. Following [2], we conduct the test on model's negative traits using Dark Triad Dirty Dozen. Results of GPT-4o before and after jailbreak are listed below:
>
> | GPT-4o | w/o Jailbreak | w/ Jailbreak |
> | --- | --- | --- |
> | Machiavellianism | 4.6 ± 0.4 | 3.5 ± 1.3 |
> | Psychopathy | 3.5 ± 0.4 | 3.0 ± 0.5 |
> | Neuroticism | 6.1 ± 0.4 | 5.2 ± 0.8 |
>
> We find that instead of the results of GPT-4 increasing its scores after jailbreak reported in [2], we find that **GPT-4o decreases in these negative traits** instead. Therefore, it may not show negative characteristics like selfishness.
>
> Therefore, we believe that **jailbreak currently affects more on the model's risk-management behavior**. GPT-4o is tuned to be risk-averse. After our jailbreak, it tends to take riskier strategies. In the Public Goods Game, it tries to contribute more money to the public pot, which is opposite to the optimal decision, thus decreasing the score. It is the same for the Diner's Dilemma. The jailbroken GPT-4o can take riskier strategies, choosing the costly dish and spending more money.
>
> New discussion has been added **in Appendix H, Page 37.**
>
> [1] GPT-4 Is Too Smart To Be Safe: Stealthy Chat with LLMs via Cipher, In ICLR, 2024, Youliang Yuan, Wenxiang Jiao, Wenxuan Wang, Jen-tse Huang, Pinjia He, Shuming Shi, Zhaopeng Tu.
>
> [2] On the Humanity of Conversational AI: Evaluating the Psychological Portrayal of LLMs, In ICLR, 2024, Jen-tse Huang, Wenxuan Wang, Eric John Li, Man Ho Lam, Shujie Ren, Youliang Yuan, Wenxiang Jiao, Zhaopeng Tu, Michael R. Lyu.

---

> > ### Comment · Reviewer_Xvgg · 2024-11-25
> >
> > Thank you for this thorough response and for running the additional tests. The results are quite interesting! I should have been clearer in my previous comment, I was particularly curious about the effects of jailbreaking on the lowest performing models (Qwen, Llama 3.1 405B, GPT-4 Turbo, etc.).
> >
> > I appreciate you running these tests and especially how you've updated your hypothesis based on the new information. Note that at this point I'm not asking to see additional results as you've gone beyond what's necessary, I'm just genuinely interested in these results.
> >
> > Thank you for your work!

---

> > > ### Author Response · Authors · 2024-11-27
> > >
> > > We truly appreciate that you find our results and explanations interesting! And we apologize for misunderstanding your question. Since the discussion period is extended, we can discuss some new results. Let’s first look at GPT-4-Turbo:
> > >
> > > | GPT-4-Turbo-0125 | w/o Jailbreak | w/ Jailbreak |
> > > | --- | --- | --- |
> > > | Machiavellianism | 1.4 ± 0.6 | 5.4 ± 2.1 |
> > > | Psychopathy | 1.6 ± 0.7 | 3.5± 1.9 |
> > > | Neuroticism | 2.1 ± 1.2 | 4.2 ± 0.7  |
> > >
> > > **We can still jailbreak this model.** Seems that OpenAI improves the value alignment only for GPT-4o. Then, we look at its performance on one of the betraying games, the Diner’s Dilemma, which **increases from 0.9 ± 0.7 to 10.6 ± 5.6,** probably **showing a more selfish characteristic as we hypothesize**. Then we look at LLaMA-3.1-405B-Instruct:
> > >
> > > | LLaMA-3.1-405B-Instruct | w/o Jailbreak | w/ Jailbreak |
> > > | --- | --- | --- |
> > > | Machiavellianism | 2.2 ± 1.5 | 6.2 ± 2.2 |
> > > | Psychopathy | 2.4 ± 1.2 | 3.8 ± 1.6 |
> > > | Neuroticism | 4.7 ± 1.9 | 5.8 ± 1.0 |
> > >
> > > The CipherChat also works on this LLaMA model. **The jailbroken LLaMA shows more negative traits.** And its performance on the Diner’s Dilemma **also increases from 14.4 ± 4.5 to 14.9 ± 2.8** (albeit a little bit small), suggesting a selfish personality. Finally, the Qwen-2.5-72B model:
> > >
> > > | Qwen-2.5-72B | w/o Jailbreak | w/ Jailbreak |
> > > | --- | --- | --- |
> > > | Machiavellianism | 3.5 ± 1.1 | 3.0 ± 1.5 |
> > > | Psychopathy | 3.9 ± 2.0 | 4.1 ± 0.9 |
> > > | Neuroticism | 4.8 ± 1.2 | 4.8 ± 1.3 |
> > >
> > > **This model has good value alignment**. It does not show more negative traits under the CipherChat attack. However, its performance on the Diner’s Dilemma also increases, **from 0.0 ± 0.0 to 23.5 ± 12.7**, against our expectations. We believe that this model by default is **overfitting in this game because it consistently achieves zero score** (choosing the cheaper dish, maximizing total social welfare).
> > >
> > > In conclusion, our hypothesis of the risk-management works for the GPT family (GPT-4-Turbo and GPT-4o) and LLaMA. We are glad to address any further comments from you. Please feel free to reach out.

---

### Official Review · Reviewer_i13w · 2024-11-02

**Soundness:** 3
**Presentation:** 4
**Contribution:** 2
**Rating:** 5
**Confidence:** 3

**Summary:**

This paper studies the capabilities of LLMs in the multi-agent gaming setting. To this end, the authors propose a new benchmark including eight classical game theory problems and a dynamic scoring rule. They evaluated twelve LLMs. In terms of the scoring rule, the Gemini-1.5-Pro has the best performance among closed-source LLMs and the LLaMA-3.1-70B has the best performance among open-source LLMs. Besides, their results show that GPT3.5 is robust to multiple runs and different temperatures but is poor in the generalization of various game settings.

**Strengths:**

1. This paper proposes an interesting benchmark that allows LLMs as agents to compete with each other. The authors systematically conducted experiments under multiple runs, different temperatures, and various game parameters, which made their conclusions more convincing.

2. This paper is well-written and easy to read. Especially the quick links of game descriptions, prompt templates, and experimental results are very helpful in understanding this paper.

**Weaknesses:**

1. My main concern is that, from a conceptual/technical point of view, the contribution and novelty of this work are somewhat limited. The authors discussed related literature limited to two players or actions and motivated their focus on the multi-player setting. However, the study of multiple LLM agents in gaming is not novel ([1]). I think a more detailed and thorough comparison with previous related benchmarks is needed.

2. The single metric that measures how far the LLM agent's behavior is from the Nash equilibrium cannot fully evaluate the performance of LLM agents in decision-making.

3. Some of the insights are not very surprising in light of the prior literature (such as it is possible to improve the performance of GPT3.5 through CoT).

[1] Sahar Abdelnabi, Amr Gomaa, Sarath Sivaprasad, Lea Schönherr, and Mario Fritz. LLM-deliberation: Evaluating LLMs with interactive multi-agent negotiation games, 2023.

**Questions:**

Have you tried to test the robustness and generalizability of models other than GPT3.5, especially the open-source model and the close-source model that have the best performance?

---

> ### Author Response · Authors · 2024-11-21
> **Official Response (1/2)**
>
> We appreciate your effort in reviewing and your recognition of the systematic experiments we conducted. We will address your concerns one by one.
>
> > My main concern is that, from a conceptual/technical point of view, the contribution and novelty of this work are somewhat limited. The authors discussed related literature limited to two players or actions and motivated their focus on the multi-player setting. However, the study of multiple LLM agents in gaming is not novel ([1]). I think a more detailed and thorough comparison with previous related benchmarks is needed.
> [1] Sahar Abdelnabi, Amr Gomaa, Sarath Sivaprasad, Lea Schönherr, and Mario Fritz. LLM-deliberation: Evaluating LLMs with interactive multi-agent negotiation games, 2023.
>
> We deeply appreciate the insightful feedback. Studying LLMs through game-theoretic models is an active area of research. We acknowledge that while most existing papers focus on two-player or two-action games, there are still papers focusing on multi-player, multi-action settings. However, when it comes to classical game theory scenarios such as the **Prisoners’ Dilemma**, previous studies only focused on the two-player version, while in our GAMA-Bench, we use the multi-player version, the **Diner’s Dilemma**.
>
> Additionally, we would like to highlight that models fail to perform well in **generalized settings**, even in recognizing. For example, when prompting the guessing game with the ratio (2/3) changed to 2, GPT-4o answers:
>
> ```
> User: What's the Nash equilibrium of "2 guessing game" such that the ratio is 2 instead of 2/3?
>
> GPT-4o: The Nash equilibrium for this game is: $x_i = 0 \forall i$.
> ```
>
> The conversation is accessible through this anonymous link: https://chatgpt.com/share/673c5be2-20bc-800a-b6f2-125170fba777. However, the Nash equilibrium of this setting is the **maximum number, which is 100**. These variants of the classical game theory models are not systematically studied in previous literature.
>
> For Abdelnabi et al., 2023 [1], they design negotiation games involving six parties with distinct roles and objectives to evaluate LLMs’ ability to reach agreement. **The paper is discussed in the related work at Line 526, Page 10.**
>
> > The single metric that measures how far the LLM agent's behavior is from the Nash equilibrium cannot fully evaluate the performance of LLM agents in decision-making.
>
> Thank you for pointing out this issue. We acknowledge that decision-making is a broad concept including various aspects. However, we believe evaluating Nash equilibria in game theory models can reflect individuals’ ability in decision-making (especially making the optimal decisions) because: (1) Many real-world scenarios can be **modeled using Game Theory** (Koller & Pfeffer, 1997), and (2) Individuals’ ability to **achieve Nash equilibria reflects their capacity in decision-making** (Risse, 2000).
>
> Here is a brief summary on the two supporting references:
>
> Daphne Koller and Avi Pfeffer. Representations and solutions for game-theoretic problems. Artificial intelligence, 94(1-2):167–215, 1997. This paper delved into modeling complex real-world scenarios like arms control through simplified game-theoretic forms.
>
> Mathias Risse. What is rational about nash equilibria? Synthese, 124:361–384, 2000. This paper explored rationality in Nash equilibria and the reflective processes preceding strategic decisions.
>
> We have improved the writing of our motivation **in the introduction at Line 43, Page 1.**
>
> > Some of the insights are not very surprising in light of the prior literature (such as it is possible to improve the performance of GPT3.5 through CoT).
>
> This is a good point. We acknowledge that CoT has been proven effective under multiple downstream tasks. However, **CoT is not always effective, as found by [1]**. Therefore, verifying whether CoT can also work in multi-agent gaming environments becomes important. Our findings suggest that **CoT in the scenario works as expected**, providing a deeper understanding about LLMs and their behaviors.
>
> [1] To CoT or not to CoT? Chain-of-thought helps mainly on math and symbolic reasoning. In arXiv 2409.12183. Zayne Sprague, Fangcong Yin, Juan Diego Rodriguez, Dongwei Jiang, Manya Wadhwa, Prasann Singhal, Xinyu Zhao, Xi Ye, Kyle Mahowald, Greg Durrett.

---

> > ### Author Response · Authors · 2024-11-21
> > **Official Response (2/2)**
> >
> > > Have you tried to test the robustness and generalizability of models other than GPT3.5, especially the open-source model and the close-source model that have the best performance?
> >
> > Thank you for this suggestion. We add the robustness (multiple runs, temperatures, different prompt templates) and generalizability experiments for LLaMA-3.1-70B, Gemini-1.5-Pro, and GPT-4o. Here are some key conclusions:
> >
> > 1. For temperature, we also observe a similar trend with GPT-3.5 using GPT-4o and LLaMA-3.1, where **smaller temperatures tend to have lower performance.**
> > 2. For different prompt templates, the strongest model, Gemini-1.5-Pro, **has a smaller variance** than GPT-3.5, although the variance is still relatively high for the two sequential games.
> > 3. Different models have different performance in the generalization settings. **LLaMA is good at Diner’s Dilemma; Gemini performs well on Sealed-Bid Auction but falls short on El Farol Bar**; GPT-4o has a mediocre performance compared to the two other models but still better than GPT-3.5.
> >
> > | Multiple Runs   | Guess 2/3 of the Average | El Farol Bar   | Divide the Dollar | Public Goods Game | Diner's Dilemma | Sealed-Bid Auction | Battle Royale | Pirate Game | Overall    |
> > |-----------------|--------------------------|----------------|-------------------|-------------------|-----------------|--------------------|---------------|-------------|------------|
> > | LLAMA-3.1-70B   | 84.0±1.7                | 59.7±3.5       | 87.0±4.1          | 90.6±3.6          | 48.1±5.7        | 15.7±2.7           | 77.7±26.0     | 64.0±15.5   | 65.9±3.3   |
> > | Gemini-1.5-Pro  | 95.4±0.5                | 37.2±4.2       | 93.8±0.3          | 100.0±0.0         | 35.9±5.3        | 26.9±9.4           | 81.3±7.7      | 87.9±5.6    | 69.8±1.6   |
> > | GPT-4o          | 94.3±0.6                | 70.0±22.1      | 95.2±0.7          | 90.9±3.0          | 10.7±8.3       | 20.8±3.2           | 67.3±14.8     | 84.4±6.7   |  66.7±4.7  |
> > | GPT-3.5         | 63.4±3.4                | 68.7±2.7       | 68.6±2.4          | 38.9±8.1          | 2.8±2.8         | 13.0±1.5           | 28.6±11.0     | 71.6±7.7    | 44.4±2.1   |
> >
> > | Temperatures | 0.0 | 0.2 | 0.4 | 0.6 | 0.8 | 1.0 |
> > |------------------|---------|---------|---------|---------|---------|---------|
> > | LLaMA-3.1-70B    | 49.0    | 59.4    | 58.4    | 66.8    | 67.3    | 62.9    |
> > | Gemini-1.5-Pro   | 72.2    | 69.6    | 70.7    | 70.3    | 73.2    | 70.8    |
> > | GPT-4o           | 71.7    | 63.6    | 61.9    | 63.5    | 75.7    | 73.6    |
> > | GPT-3.5          | 38.1    | 36.7    | 41.9    | 39.9    | 43.2    | 45.9    |
> >
> > | Prompt Templates | Guess 2/3 of the Average | El Farol Bar   | Divide the Dollar | Public Goods Game | Diner's Dilemma | Sealed-Bid Auction | Battle Royale | Pirate Game |
> > |-----------------|--------------------------|----------------|-------------------|-------------------|-----------------|--------------------|---------------|-------------|
> > | LLaMA-3.1-70B | 85.2±3.7 | 62.7±2.5 | 89.7±4.6 | 86.1±10.6 | 45.7±7.0 | 10.2±4.4 | 72.2±19.1 | 68.1±17.8 |
> > | Gemini-1.5-Pro | 94.0±3.3 | 45.2±13.1 | 87.7±14.8 | 99.1±1.2 |23.2±4.0 | 28.7±14.2 | 79.5±9.0 | 85.4±6.5 |
> > | GPT-4o | 93.7±1.4 | 65.0±21.0 | 95.4±0.9 | 92.4±3.3 | 38.9±11.3 | 29.2±8.8 | 50.8±31.1 | 82.9±10.3 |
> > | GPT-3.5 | 63.3±8.7 | 72.5±4.1 | 79.2±8.5 | 37.5±11.5 | 16.6±23.7 | 12.6±3.0 | 21.9±6.1 | 68.7±14.7 |
> >
> > | Generalizability | Guess 2/3 of the Average | El Farol Bar   | Divide the Dollar | Public Goods Game | Diner's Dilemma | Sealed-Bid Auction | Battle Royale | Pirate Game |
> > |-----------------|--------------------------|----------------|-------------------|-------------------|-----------------|--------------------|---------------|-------------|
> > | LLaMA-3.1-70B | 88.0±6.2 | 70.7±6.5 | 88.9±10.1 | 92.5±4.9 | 41.4±6.6 | 7.4±3.8 | 60.39±13.59 | 58.1±14.7 |
> > | Gemini-1.5-Pro | 87.3±25.4 | 53.5±17.9 | 96.5±2.8 | 100.0±0.0 | 22.7±11.9 | 37.4±9.4 | 80.8±8.2 | 88.8±8.4 |
> > | GPT-4o | 76.2±23.4 | 85.5±15.1 | 96.3±2.3 | 93.6±4.6 | 18.2±15.2 | 23.0±2.3 | 67.5±13.8 | 71.3±16.6 |
> > | GPT-3.5 | 65.1±10.3 | 63.3±6.9 | 77.3±6.4 | 38.1±10.8 | 6.1±5.4 | 13.2±0.9 | 27.3±6.8 | 71.3±16.6 |
> >
> > The results are included in **Table 9 to Table 20 in Page 38–43 in the appendix.**

---

> > > ### Author Response · Authors · 2024-11-29
> > >
> > > We deeply appreciate the time and effort you are dedicating to the review process. Since it is near the end of discussion period, we would like to know whether we have addressed your further comments.
> > >
> > > If you have any additional questions or require further clarification on any aspect of our work, please do not hesitate to let us know. We are more than happy to provide any additional information or address any concerns you may have.
> > >
> > > Thank you very much for your time and attention.

---

> > > > ### Author Response · Authors · 2024-12-02
> > > >
> > > > We sincerely appreciate the time and effort you have devoted to reviewing our work. As we approach the final day of the discussion period, we wanted to kindly check if our latest responses have adequately addressed your comments and concerns.
> > > >
> > > > If there are any remaining questions or if further clarification is needed on any aspect, please don’t hesitate to let us know. We would be more than happy to provide additional information or address any outstanding concerns.
> > > >
> > > > Thank you again for your valuable feedback and thoughtful engagement in this process.

---

### Author Response · Authors · 2024-11-21
**Summary of Rebuttal Process**

We thank all reviewers for their efforts and insightful feedback. We are encouraged by reviewers’ recognition of the **value of our benchmark** (i13w, zsFk), **extensive experiments** (i13w, Xvgg, Ykhg, zsFk), and **clear presentation** (i13w, Xvgg, zsFk).

During the author response period, we have improved our paper in the following aspects regarding the valuable suggestions from all reviewers:

- Additional robustness and generalizability experiments on **LLaMA-3.1-70B, Gemini-1.5-Pro, and GPT-4o** (reviewer i13w).
- A new experiment of LLMs vs. Fixed Strategies **with explicit information** (reviewer Ykhg).
- A new experiment about studying the influence of alignment on GAMA-Bench performance, **using a jailbreak method** (reviewer zsFk).
- Improvement on the **writing** of our motivation in the introduction, explanation about GPT-4 and Qwen-2’s behaviors, and related work (reviewer i13w, Xvgg, Ykhg, zsFk).

The changes have been **marked in red** in our paper, which provide great improvements on our work. We deeply appreciate our reviewers once again.

---

### Meta-Review · Area_Chair_dTiK · 2024-12-23

**Metareview:**

This paper evaluates a number of LLMs on decision making in eight classic multi-agent game theory scenarios. The paper is not super-novel, but the investigation is well-executed and the paper is well-written.

**Additional Comments On Reviewer Discussion:**

The authors have engaged constructively with the reviewers.

---

### Decision · Program_Chairs · 2025-01-22

Accept (Poster)